# CD177 modulates the function and homeostasis of tumor-infiltrating regulatory T cells

Myung-Chul Kim[1,2,26], Nicholas Borcherding[3,4,5,26], Kawther K. Ahmed [3,24,26], Andrew P. Voigt [5,26], Ajaykumar Vishwakarma[4,6], Ryan Kolb[1,2], Paige N. Kluz[3], Gaurav Pandey[3,25], Umasankar De[1,2], Theodore Drashansky [7], Eric Y. Helm[7], Xin Zhang [2,8], Katherine N. Gibson-Corley[9], Julia Klesney-Tait[10], Yuwen Zhu [11], Jinglu Lu [12], Jinsong Lu[13], Xian Huang[13], Hongrui Xiang[13], Jinke Cheng [14,15], Dongyang Wang[16], Zheng Wang[16], Jian Tang[16], Jiajia Hu[17], Zhengting Wang[18], Hua Liu[18], Mingjia Li[19], Haoyang Zhuang[19], Dorina Avram[2,7,20], Daohong Zhou [2,8], Rhonda Bacher [21], Song Guo Zheng[22], Xuefeng Wu [12,13,14,23✉], Yousef Zakharia [9✉] & Weizhou Zhang [1,2,3✉]

Regulatory T (Treg) cells are one of the major immunosuppressive cell types in cancer and a potential target for immunotherapy, but targeting tumor-infiltrating (TI) Treg cells has been challenging. Here, using single-cell RNA sequencing of immune cells from renal clear cell carcinoma (ccRCC) patients, we identify two distinct transcriptional fates for TI Treg cells, Fate-1 and Fate-2. The Fate-1 signature is associated with a poorer prognosis in ccRCC and several other solid cancers. CD177, a cell surface protein normally expressed on neutrophil, is specifically expressed on Fate-1 TI Treg cells in several solid cancer types, but not on other TI or peripheral Treg cells. Mechanistically, blocking CD177 reduces the suppressive activity of Treg cells in vitro, while Treg-specific deletion of *Cd177* leads to decreased tumor growth and reduced TI Treg frequency in mice. Our results thus uncover a functional CD177[+] TI Treg population that may serve as a target for TI Treg-specific immunotherapy.

---

A list of author affiliations appears at the end of the paper.

Treg cells are a population of T cells with suppressive effects on a variety of immune cells including CD8[+] T cells, CD4[+] T cells, natural killer cells, and dendritic cells[1]. Treg cells play an indispensable role in maintaining normal immune homeostasis and peripheral tolerance. Their suppressive activity in the tumor microenvironment is associated with the loss of anti-tumor immunity[1], which provides the rationale for the development of Treg-targeted immunotherapy[2]. For example, ipilimumab is the first immune checkpoint blocker approved by the Food and Drug Administration and has been thought to work through inhibiting CTLA-4 on effector T cells as well as on Treg cells by depleting Treg cells in the tumor microenvironment[3,4]. Treg cells are identified by the expression of transcription factor forkhead box P3 (FOXP3) that acts as a master regulator for Treg development and suppressive function[5,6]. FOXP3 alone fails to predict patient survival and its prognostic value is tumor type- and stage-dependent across different cancers[7,8]. Recent studies also unveiled functional heterogeneity of FOXP3[+] Treg cells in peripheral blood (PB)[9], as well as in different tumor types including colorectal cancer[10] and glioma[11]. Disruption of all Treg function may negatively influence immune homeostasis, as seen in the autoimmune complications associated with CTLA-4 blockade, which underscores the need for more refined targets for TI-Treg cells.

Recent RNA sequencing of blood-derived, normal-tissue-resident, and TI Treg cells in two independent studies revealed the transcriptional overlap between normal-tissue and TI Treg cells[12,13]. TI Treg cells showed differential upregulation of appreciable number of genes, including chemokine receptor 8 (CCR8), CTLA4, lymphocyte activation gene 3 (LAG3), and T-cell membrane protein 3 (TIM-3, encoded by HAVCR2), layilin (LAYN), and MAGE family member H1 (MAGEH1)[12,13]. One report from the Rudensky group identified CD177[+] TI Treg cells in breast cancer with no further functional elaboration[12]. A comprehensive single cell RNA sequencing using mouse splenocytes also defines the expression of Cd177 in a small subset of mouse splenic Treg cells[14]. CD177 (also known as NB1, HNA-2a, or PRV1) is a glycosylphosphatidylinositol-linked cell surface protein that is expressed heterogeneously by neutrophils and has been identified as a useful biomarker for myeloproliferative diseases[15,16]. We recently generated a Cd177 knockout (KO) mouse model and demonstrated a potential role of CD177 in neutrophil viability[17]. Limited literature indicates a correlation between the loss of CD177 expression and poor prognosis in colorectal and gastric cancer[18,19]. Our recent study demonstrated that CD177 is expressed by epithelial cells of breast cancer where its expression is associated with a better prognosis. We further found that CD177 expressed by cancer cells has tumor-suppressive functions via regulating β-catenin activation[20].

To investigate transcriptional heterogeneity of TI Treg cells, we performed single-cell RNA-sequencing of PB and TI immune cells in three ccRCC patients[21]. We chose ccRCC based on the responsiveness of these tumors to immune checkpoint inhibitors (ICI) in spite of their low mutational load, which implies the importance of the tumor microenvironment in mediating the sensitivity to ICI[22]. It has been shown that Treg gene signatures are associated with a worse prognosis in ccRCC[23,24] and Treg infiltration into ccRCC tumors has been implicated in poor response to IL-2-based immunotherapy and targeted therapy[25,26].

Here we demonstrate, by pairing our dataset with the single-cell sequencing of immune cells from an HCC dataset[27], the presence of transcriptionally and functionally distinct TI Treg populations. Most importantly, several genes, including BCL2L1[28] and CD177, are highly elevated in TI Treg cells relative to other TI T cells or PB Treg cells. Our data thus support that CD177 is specifically expressed within a specific population of TI Treg cells

of solid cancers, and that CD177 is able to mediate the suppressive activity of TI Treg cells in cancer.

## Results

### Analysis of TI Treg cells using single-cell RNA sequencing from ccRCC patients
Using our newly developed ccRCC single cell RNA sequencing dataset including 13,433 PB and 12,239 TI cells, we identified a cluster of Treg cells based on the expression of FOXP3 and CD25 (IL2RA) (Fig. 1a, b), containing 160 PB and 574 TI Treg cells. There were 273 differentially-expressed genes (DEGs) (Log fold-change > 1, adjusted p-value < 0.05) by comparing TI versus PB Treg cells. We further filtered the DEGs by comparing the percent of TI and PB Treg cells that express each gene (calculating the Δ percentage difference) (Fig. 1c) and a complete analysis is available in Supplementary Data 1. The upregulated DEGs in TI Treg cells with Δ percentage difference > 20% were labeled (Fig. 1c). Of those genes, two genes were only expressed in TI Treg cells and had 0% expression in PB Treg cells, i.e. NR4A1 (Δ percentage difference = 51.4%) and CD177 (Δ percentage difference = 20.2%) (Fig. 1c). A summary of the top eight upregulated or downregulated DEGs within TI Treg cells was shown (Fig. 1d, e).

We also analyzed a recently-published single-cell profile of T cells in HCC[27], comprised of flow-sorted T cells including Treg cells from PB, normal liver parenchyma, a transitional/junctional zone near the tumor, and TI Treg cells. After clustering of Treg cells (Supplementary Fig. 1a, b), we performed a similar DEG analysis comparing 634 TI versus 264 PB Treg cells (Supplementary Fig. 1c). We found a total of 467 DEGs in the HCC TI Treg cells (a complete analysis is available in Supplementary Data 2), among which there were 143 shared DEGs with the ccRCC TI Treg cells (Fig. 1f). The overlapping coefficient was 0.52, reflecting the significant effect of the tumor microenvironment and patient heterogeneity on the gene expression profile of TI-Treg cells as reported before[26,29]. In agreement with previous reports, we also found the increased expression of CCR8, LAYN, and MAGEH1 in TI Treg cells[12,13] (Fig. 1g). Among the shared 143 DEGs, CD177 has the highest average log-fold change in ccRCC (log-fold change = 4.86) and HCC (log-fold change = 4.55) from TI Treg cells (Fig. 1g). Several other Treg relevant genes, including CTLA4, ICOS, NR4A1 (NUR77), TNFRSF18 (GITR), TNFRSF4 (OX40), and TNFRSF9 (4-1BB), were also differentially upregulated in TI Treg cells (Fig. 1g). A similar set of upregulated genes was seen in the RNA sequencing result from pooled Treg cells in breast, colorectal, and lung cancers (Supplementary Fig. 1d, e).

### The transcriptional bifurcation of TI Treg cells
Unlike previous studies that relied on pooled TI Treg cells and PB Treg cells, we were able to investigate the dynamic processes of Treg transcriptomes at the single cell level (Fig. 2). Using the reverse graph embedding approach in the Monocle 2 algorithm[30], we constructed a cell trajectory of the TI and PB Treg cells in ccRCC (Fig. 2a) and HCC (Supplementary Fig. 2a) datasets. We studied the gene expression in Treg cells along differentiation processes (Fig. 2a, b). To understand the process represented by an estimated trajectory, we first identified genes that were significantly dynamic along pseudotime by fitting a negative binomial generalized additive model (GAM) of gene expression as a smooth function of pseudotime and the significance was assessed using likelihood ratio testing[30]. Many dynamic genes that were increased in their expression within TI Treg cells along pseudotime were previously identified in the differential expression analysis such as CCR8, CD177, CCL20, and CTLA4 (Fig. 2b), with the complete list available in Supplementary Data 3. A similar

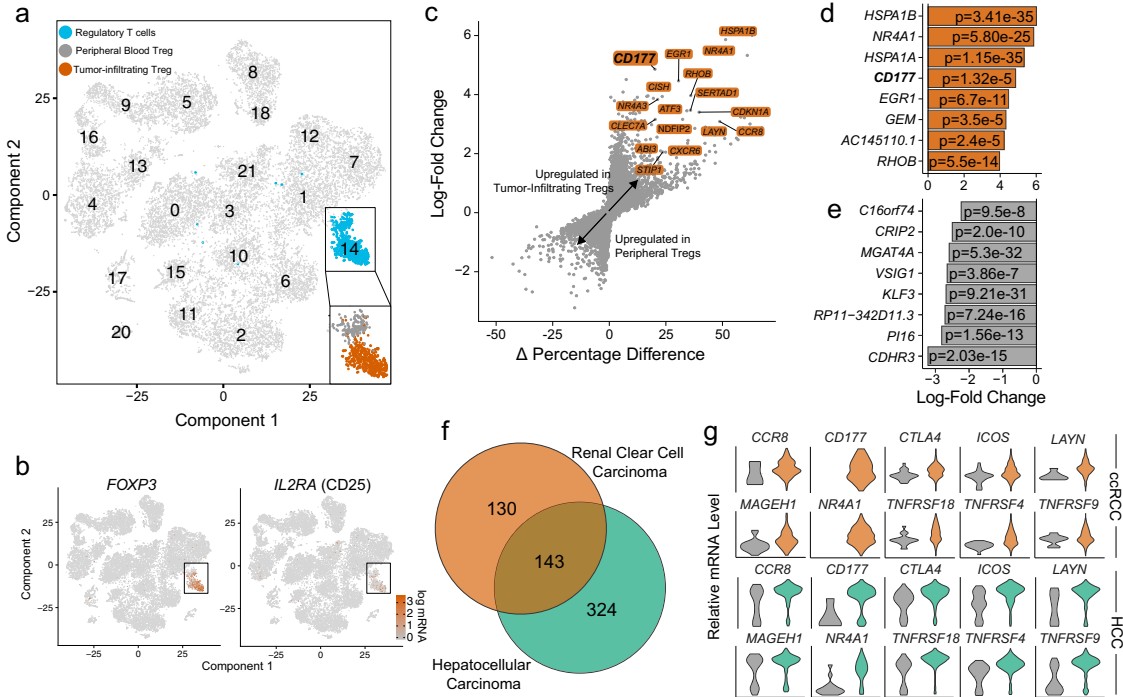

**Fig. 1 TI Treg cells display a distinct expression program compared to PB controls in the ccRCC single cell RNA sequencing dataset. a** tSNE projection of immune cells from three ccRCC patients with normal PB cells ($n = 13,433$) and TI cells ($n = 12,239$). Treg population (blue) was isolated and separated as TI (orange) *versus* PB Treg cells (gray). **b** tSNE projection with the highlighted expression of Treg markers, *FOXP3* and *IL2RA* (CD25). **c** Differential gene expression analysis using the log-fold change expression versus the difference in the percentage of cells expressing the gene comparing TI versus PB Treg cells (Δ Percentage Difference). Genes labeled have log-fold change > 1, Δ Percentage Difference > 20% and adjusted *P*-value from Wilcoxon rank sum test <0.05. **d** Top eight upregulated genes by log-fold change in TI Treg cells with adjusted *P*-value <0.05. **e** Top eight downregulated genes by log-fold change in TI Treg cells with adjusted *P*-value <0.05. **d, e** Wilcoxon rank sum test with p-values adjusted using the Bonferroni method. **f** Comparison of differential genes in TI Treg cells in ccRCC (orange) and HCC (green) compared to PB Treg cells. Significant genes were defined as log-fold change >1 or < −1 with adjusted *P*-values <0.05. **g** Violin plots showing relative mRNA level of Treg markers in PB (gray) and TI Treg cells in ccRCC (top) and HCC (bottom).

profile of genes was expressed dynamically along pseudotime in the HCC Treg cells, although with reduced separation between TI and PB Treg cells in several genes including *CTLA4* and *CCR8* (Supplementary Fig. 2b). In contrast, other genes such as *CCL20*, *CCL4* and *CD177* maintained the differential expression levels between TI and PB Treg cells within the HCC dataset (Supplementary Fig. 2b). We observed a significant increase in chemokines *CCL20* and *CCL4* over pseudotime trajectory (Fig. 2b, Supplementary Fig. 2b). These chemokines are reported to play a role in the trafficking of Treg cells to the site of antigen presentation[31].

The trajectory from PB to TI Treg cells contained two distinct branches (Fig. 2a) that were not detectable using Treg marker genes, nor were they revealed in the previous clustering analysis (Fig. 1a). To study the two branches of TI Treg cells, we performed a branched expression analysis modeling by including branch identity as a covariate in the full GAM model for the likelihood ratio test (Fig. 2c). Commonly associated genes of immune regulation and suppression were increased in their expression in TI Treg cells of Cell Fate #1 (CF1) compared to the Treg cells of Cell Fate #2 (CF2) (Fig. 2c). The CF2 Treg cells had increased expression of ribosomal-associated genes (Fig. 2c). Using the differential gene analysis, we were able to distinguish three distinct trends in gene expression between the two TI Treg fates: 1) non-specific increase across the two TI Treg fates, like *CCR8* and *CTLA4*; 2) increased in the CF1 only, like *CD177* and *TNFRSF4* (OX40); and 3) increased in the CF2 only, like *CXCR4* and *EGR1* (Fig. 2d). To investigate the potential functional difference between the two cell fates, we performed single-sample

gene set enrichment analysis (GSEA) using gene sets for T cell phenotypes and for comparing single-cell gene expression at the poles of the trajectory (Fig. 2e). Both TI Treg cell fates had a reduction in the naïve T cell signature and an increase in T cell exhaustion signature compared to PB Treg cells in ccRCC (Fig. 2e, upper row) and HCC (Supplementary Fig. 2d). In contrast, the cytotoxicity and cell cycle signatures were significantly higher in CF1 Treg cells (Fig. 2e, lower row), along with an increase of the percent of cells in the $G_2M$ phase (Fig. 2f). The increase in $G_2M$ was not seen in the CF1 Treg cells in the HCC dataset (Supplementary Fig. 2e), suggesting a possible role of tumor microenvironment in the cell cycle regulation. We also observed a clonal expansion in TI Treg cells across all ccRCC patients (Supplementary Fig. 3a–c), in agreement with the published observation[12,27,32,33]. The clonal expansion of T cell receptor repertoire was enriched in the CF1 TI Treg cells (Supplementary Fig. 3d). This supports the previous observation of increased clonality within the CD177[+] TI Treg cells[12].

**The suppressive TI Treg subset expresses a gene signature with superior prognostic ability.** As our data suggests transcriptional and functional differences in a suppressive subset of TI Treg cells, we investigated whether a gene signature developed from single-cell RNA sequencing data would provide improved prognostic ability (Fig. 3a) by evaluating the Kidney Renal Clear Cell (KIRC) dataset from the Cancer Genome Atlas (TCGA). The KIRC dataset contains 533 primary tumor samples with overall survival information. We performed feature selection of randomly

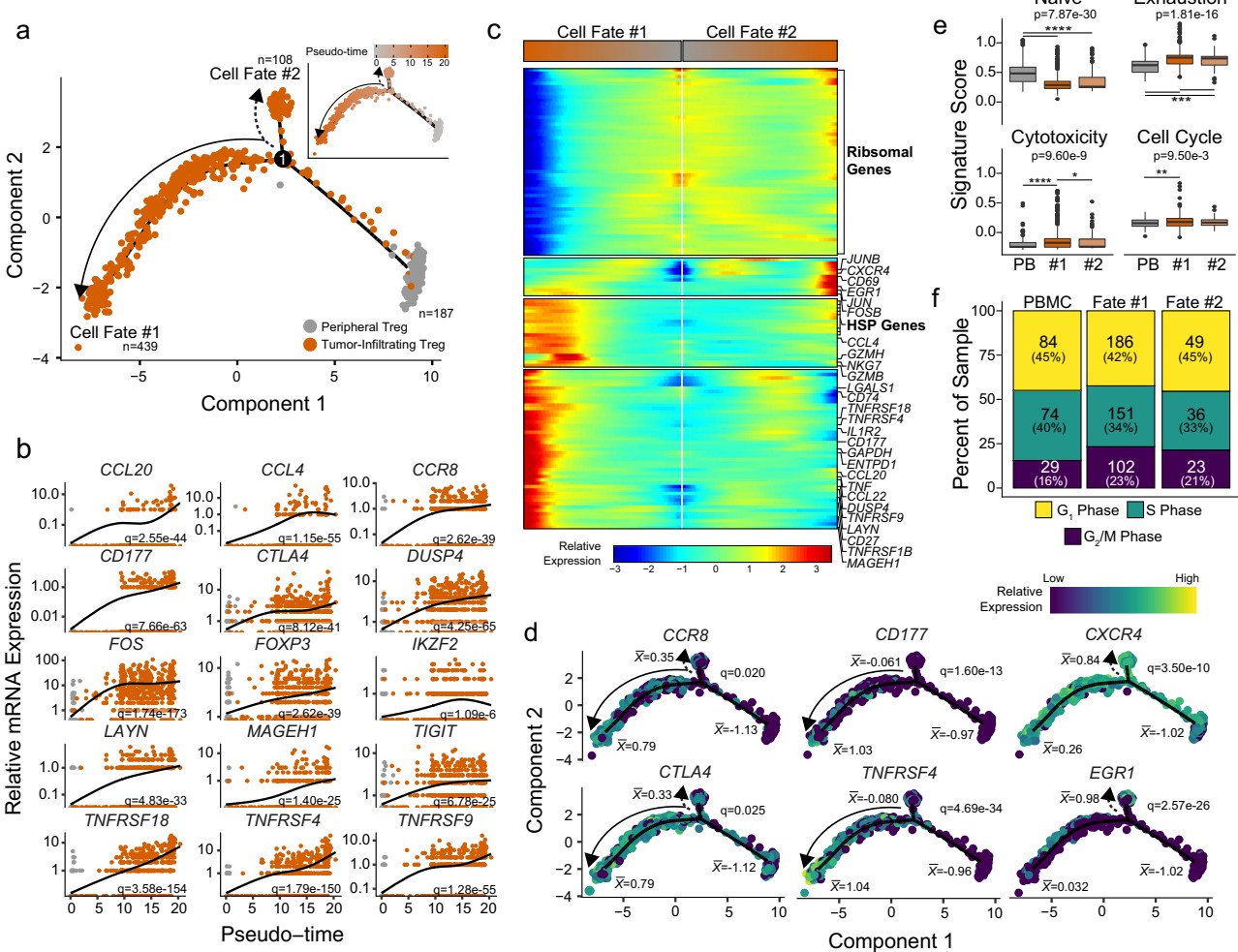

**Fig. 2 Bifurcation in the transcriptional state of TI Treg cells reveals a more suppressive cell fate. a** Trajectory manifold of Treg cells from the ccRCC using the Monocle 2 algorithm. Solid and dotted lines represent distinct cell trajectories/fates defined by expression profiles. **b** Pseudotime projections of transcriptional changes in immune genes based on the manifold. The significance was determined based on differential testing relative to the site of origin which was also used to generate pseudotime and adjusted for multiple comparisons. **c** Expression heatmap of significant (q < 1e-6) genes based on branch expression analysis comparing the two TI cell fates. The genes in the heatmap were also used in the ordering of the pseudotime variable. **d** Cell trajectory projections of transcriptional changes in immune genes based on the manifold. Significance based on differential testing between the first and second cell fates of TI Treg cells. $\bar{x}$ denotes the scaled mean mRNA levels at each pole of the manifold. **e** Gene set enrichment analysis of the poles of the trajectory manifold. Boxplots were drawn for values between 25th and 75th percentile with median value lines. Outlier values were graphed as individual points for values 1.5 times the interquartile range. *P*-values are based on one-way ANOVA with individual comparisons corrected for multiple hypothesis testing using the Tukey HSD method. **f** Results of the cell cycle regression analysis of single cells for each cell fate using the Seurat R package.

selected 10% training KIRC cohort samples to identify gene signatures using three gene sets from the single-cell analysis: (1) 143 differential genes of TI Treg cells shared between ccRCC and HCC, (2) 86 genes differentially expressed in the CF2, and (3) 222 genes differentially expressed in the CF1. These genes were passed through a gene filter for genes specific to TI Treg cells compared to other TI-immune cells derived from the ccRCC single-cell cohort (Supplementary Data 4). After evaluating polynomial support vector machines, k-nearest neighbors, boosted tree classifications, and bootstrap aggregating classifications supervised machine-learning models, we selected a linear support vector machine because the method had the least issue in overfitting the survival data. The output of this process led to the selection of six Treg-specific genes (Fig. 3a). Applying these signatures to the remaining 90% of the KIRC samples, we found that the TI-Treg signature (Fig. 3b, upper panel) and the CF1 signature (Fig. 3b, lower panel) could discriminate between patients with better and poorer overall prognosis, but not the CF2 signature. Moreover,

both the CF1 and total TI Treg signatures had superior discrimination ability when compared to previously reported tumor Treg markers, such as *FOXP3* and ratio-based signatures *CCR8:FOXP3* or *CCR8:CD3G* (Fig. 3c)[12,13].

Although the TI-Treg signature had a larger hazard ratio compared to the CF1 signature in KIRC (Fig. 3c), the CF1 signature was superior in discriminating patient survival within multiple cancer types (Fig. 3d, e). We applied the ccRCC-signatures across the 24 largest TCGA datasets, finding that the TI Treg signature was only able to predict poor survival within KIRC and liver hepatocellular carcinoma (LIHC) (Fig. 3d), the cancer types from which the common TI Treg signature was derived. The CF2 signature failed to discriminate groups based on overall survival in any TCGA dataset. The CF1 TI Treg signature was able to predict poor survival within KIRC and LIHC, as well as 5 other cancers (Fig. 3e). Both CF1 and the common TI Treg gene signatures discriminated overall survival for histological grade 3 samples for KIRC, but not other grades, reducing the

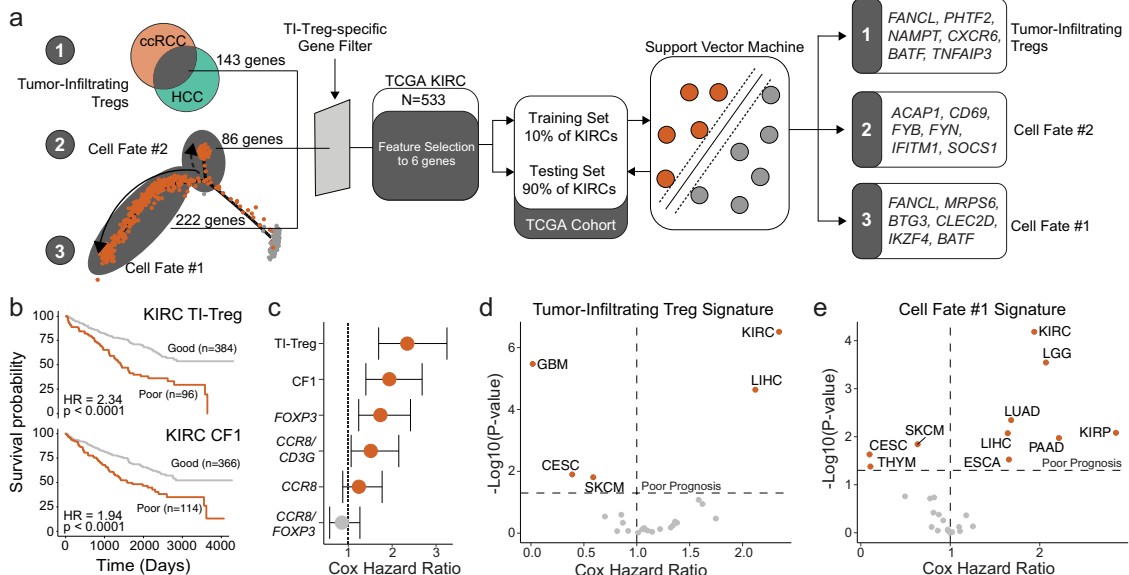

**Fig. 3 Improved prognostic prediction is associated with a signature from the suppressive TI Treg cell fate. a** Schematic of signature development using feature selection from: (1) 143 common differential genes of TI Treg in ccRCC and HCC, (2) 86 genes differentially expressed in CF2, and (3) 222 genes differentially expressed in CF1 using the 10% of the TCGA KIRC/ccRCC dataset for feature selection. Gene signatures generated after feature selection were used to predict prognosis in the remaining 90% TCGA KIRC/ccRCC, as well as 23 other TCGA cancer datasets. **b** Kaplan–Meier curves for overall survival in TCGA KIRC/ccRCC using the TI-Treg common gene signature (upper panel) and CF1-Treg gene signature (lower panel). **c** Prognostic prediction for Treg signatures compared to other proposed signatures for TI Treg cells. Hazard ratios with the bars representing the 95% confidence intervals, and *P* values derived from Cox proportional hazard regression modeling. **d, e** Overall survival prediction with Cox proportional hazard ratio and −log10(*P* value) based on two-sided log-rank testing across the 24 largest TCGA datasets using **d** the TI Treg signature and **e** the CF1 signature. CESC, Cervical squamous cell carcinoma and endocervical adenocarcinoma; ESCA, esophageal carcinoma; GBM, glioblastoma multiforme; KIRC, kidney renal clear cell carcinoma, KIRP, kidney renal papillary cell carcinoma; LGG, low-grade glioma; LIHC, liver hepatocellular carcinoma; LUAD, lung adenocarcinoma; PAAD, pancreatic adenocarcinoma; SKCM, skin-cutaneous melanoma; THYM, thymoma.

concern for selection bias towards more advanced histological grades (Supplementary Fig. 3e).

**Expression of CD177 is specific to TI-Treg cells in various human solid cancers.** We became particularly interested in CD177 due to the increased expression in TI Treg cells and its specificity within the CF1 Treg cells in ccRCC (Fig. 4a, b) and HCC (Supplementary Fig. 4a–c). We developed an immunohistochemistry (IHC) protocol and validated that CD177 protein was expressed in neutrophils with polymorphic nuclei from human colon tissue (Supplementary Fig. 4d, lower panel). In addition, we found that CD177 protein was expressed by epithelial cells from the colon, breast, and prostate (Supplementary Fig. 4e, black arrows), as well as immune cell infiltrates from various human tissues including colon, breast, prostate, liver, lung, lymph node and spleen (Supplementary Fig. 4e, brown arrows). We did observe $CD177^+$ Treg cells within the normal liver and transitional/junctional zones of HCC from the single cell data (Supplementary Fig. 4c), as well as $CD177^+$ Treg cells from human splenocytes of cancer patients (Supplementary Fig. 4f). A comprehensive analysis of all T cell subtypes within the ccRCC dataset further supports the exclusive expression pattern of $CD177$ on TI Treg cells within tumors, but not on other T cell subtypes regardless of effector/memory phenotypes of $CD4^+$ or $CD8^+$ T cells (Fig. 4c). We performed flow cytometry on TI lymphocytes and demonstrated that only TI Treg cells expressed various levels of CD177 (Fig. 4d, Supplementary Fig. 4g, h), but neither PB Treg cells, non-Treg CD4 T (Tconv) cells, nor CD8 T cells expressed significant amount of CD177 (Fig. 4d, Supplementary Fig. 4g, h). There was an average of 22.4% $CD177^+$ Treg cells among TI Treg

cells in breast cancer and of 16.8% in ccRCC (Fig. 4e). There was a negligible expression of CD177 on conventional $CD4^+$ T cells (Tconv) in breast and renal cancers (Fig. 4e). Using dual-color IHC staining, we identified CD177- and FOXP3-double-positive cells in breast cancer sections (Supplementary Fig. 4i, red arrow). To identify factors that can induce CD177 expression on Treg cells, we purified human PB Treg cells or CD4 Tconv cells (Fig. 4f) from breast cancer patients, or splenic Treg cells or CD4 Tconv from MC38 tumor-bearing mice (Fig. 4g), either non-stimulated (ns) or treated with anti-CD3/CD28 and IL-2. We failed to induce CD177 expression within human and mouse Treg cells, human $CD44^+CD45RA^-$ and $CD44^+CD45RA^+$ CD4 Tconv cells, or mouse $CD44^+CD62L^-$ and $CD44^-CD62L^-$ CD4 Tconv cells after 4 days of induction (Fig. 4f, g).

Using the germline $Cd177$-KO mice[17], we found that $Cd177$-deficiency led to a significant decrease in tumor growth from the Py8119 orthotopic breast cancer model (Fig. 5a, female mice) or the MC38 colon cancer model (Fig. 5b, male mice). We further developed a novel mouse model carrying a floxed allele of $Cd177$ ($Cd177^{fl/fl}$) (Supplementary Fig. 5), which was crossed with the $Foxp3^{YFP/Cre}$ mice to generate mice with Treg-specific deletion of $Cd177$. We transplanted MC38 cells either into $Cd177^{fl/fl}$ or $Cd177^{fl/+}/Foxp3^{YFP/Cre}$ males (collectively referred to as the control group), or age-matched littermates of $Cd177^{fl/fl}/Foxp3^{YFP/Cre}$ males. Treg-specific $Cd177$ deletion resulted in reduced tumor growth (Fig. 5c), similarly as in the germline $Cd177$ KO mice (Fig. 5b). The CD177 KO was validated by real-time PCR for germline KO strain (Supplementary Fig. 6a). Flow cytometry showed significant decrease in CD177 protein expression within

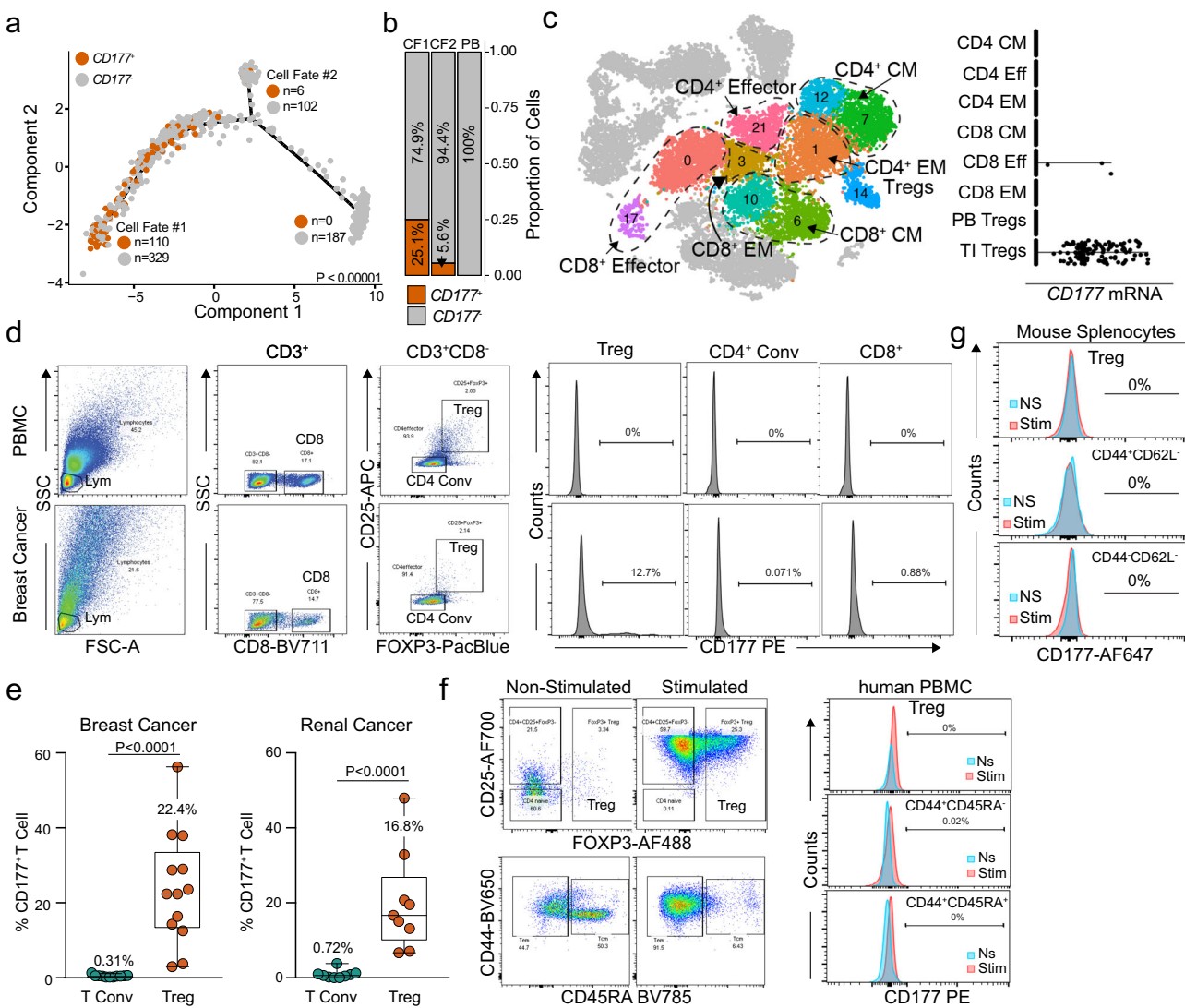

**Fig. 4 CD177 is a marker for a subpopulation of TI Treg cells. a** Trajectory manifold of Treg cells from the ccRCC TI Treg cells with the number of *CD177+* and *CD177−* Treg cells for each respective cell fate. The significance is based on $\chi^2$ testing comparing the three poles of the manifold. **b** Proportional distribution of *CD177+* Treg cells by cell fate across the manifold. **c** *CD177* mRNA expression on different effector/memory T cell subpopulations within ccRCC dataset defined by a gene signature including *CD27, CD28, CCR7, CCR5, SELL* and *FAS*. CM: central memory; Eff, effector; EM, effector memory. **d** Schematic flow cytometry data gating on lymphocytes (lym) were further analyzed for CD177 expression on Treg cells (CD4+CD25+FOXP3+), conventional CD4 T cells (CD4 conv, CD4+CD25−) and CD8 T cells (CD8) isolated from TI lymphocytes in breast cancer or PBMC. **e** Percent CD177+ cells within TI CD4 Tconv or TI Treg cells in breast cancer ($n = 13$, mean ± SD) and renal cancer ($n = 9$, mean ± SD). Two-sided unpaired T-test was used. **f** CD177 protein expression on different T cell subpopulations defined by CD44 and CD45RA from PBMC of breast cancer patients. Treg cells (CD4+CD25+CD127low) and CD4 conv cells (CD4 conv, CD4+CD25−) were purified from human PBMC of breast cancer patients, either non-stimulated or stimulated (anti-CD3/CD28 + IL-2). Left: gating of Treg cells (top) or effector/memory population for CD4 Tconv (bottom); Right: CD177 expression on the gated populations. **g** CD177 protein expression on different effector/memory T cell subpopulations defined by CD44 and CD62L from mouse spleens. Treg cells (CD4+CD25+GITR+) and CD4 conv cells (Tconv, CD4+CD25−) were purified from spleens of WT mouse, either non-stimulated or stimulated (anti-CD3/CD28 + IL-2). **f, g** CD177 expression was determined by flow cytometry.

CD25+FoxP3+ TI Treg cells from the Treg-specific CD177 KO mice (Supplementary Fig. 6b).

To determine the influence of Treg-specific KO of *Cd177*, we performed a thorough immune profiling from different tissues of normal (non-tumor bearing) or tumor-bearing mice using flow cytometry (Supplementary Fig. 6c, gating strategy for lymphocytes). CD177 protein was detectable on Treg cells from various tissues – including tumor, lung, spleen, and thymus of WT tumor-bearing mice with the highest expression from TI Treg cells (Supplementary Fig. 6d, upper panels). CD177 protein was

undetectable on CD4+ Tconv or CD8+ T cells except that TI CD8 T cells have a low level of CD177 expression (Supplementary Fig. 6d, upper panels), a phenotype that is different from human TI CD8+ T cells (Fig. 4d, Supplementary Fig. 4h). CD177 protein was largely diminished on Treg cells from these tissues of mice carrying Treg-specific KO of *Cd177* (Supplementary Fig. 6d, lower panels). Neutrophils from these tumors or lungs, however, exhibited similar levels of CD177 (Supplementary Fig. 6e). While we did not observe significant alterations within immune populations from non-tumor-bearing mice (Supplementary

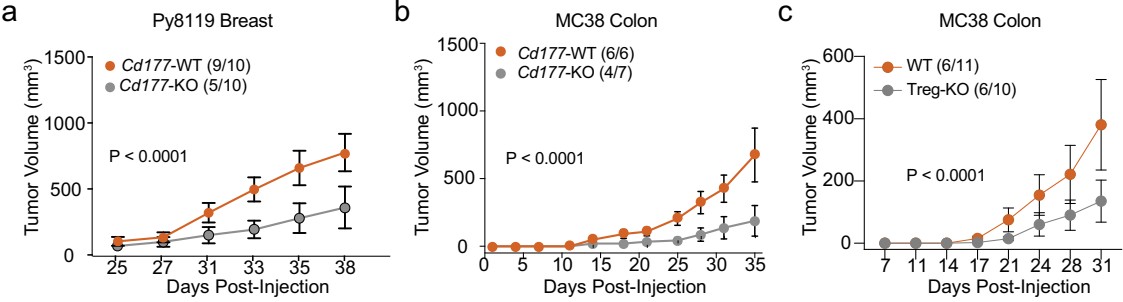

**Fig. 5 *Cd177*-deficiency in Treg cells leads to reduced tumorigenesis. a** Py8119 tumor growth is significantly reduced in *Cd177*-KO mice compared to WT, $P < 0.0001$ in female mice challenged with $5 \times 10^2$ cells per inoculation, $n = 10$ bilateral tumors. **b** MC38 tumor growth is significantly reduced in *Cd177*-KO mice compared to WT, $P < 0.0001$ in male mice challenged with $5 \times 10^4$ cells per inoculation ($n = 6$ WT and 7 KO). **c** MC38 tumor growth is significantly reduced in Treg-specific *Cd177*-KO (*Cd177^{fl/fl}/Foxp3-Cre*) male mice compared to control male mice either carrying floxed *Cd177* allele (*Cd177^{fl/fl}*) or *Cd177^{fl/+}/Foxp3-Cre*. $P < 0.0001$ in mice challenged with $5 \times 10^4$ cells per inoculation ($n = 10$ WT and 11 Treg-KO). **a–c** Numbers in parenthesis equates to the number of mice that developed palpable tumors/total mice inoculated. Data are presented as mean ± SEM. Two-way ANOVA was used for **a–c**.

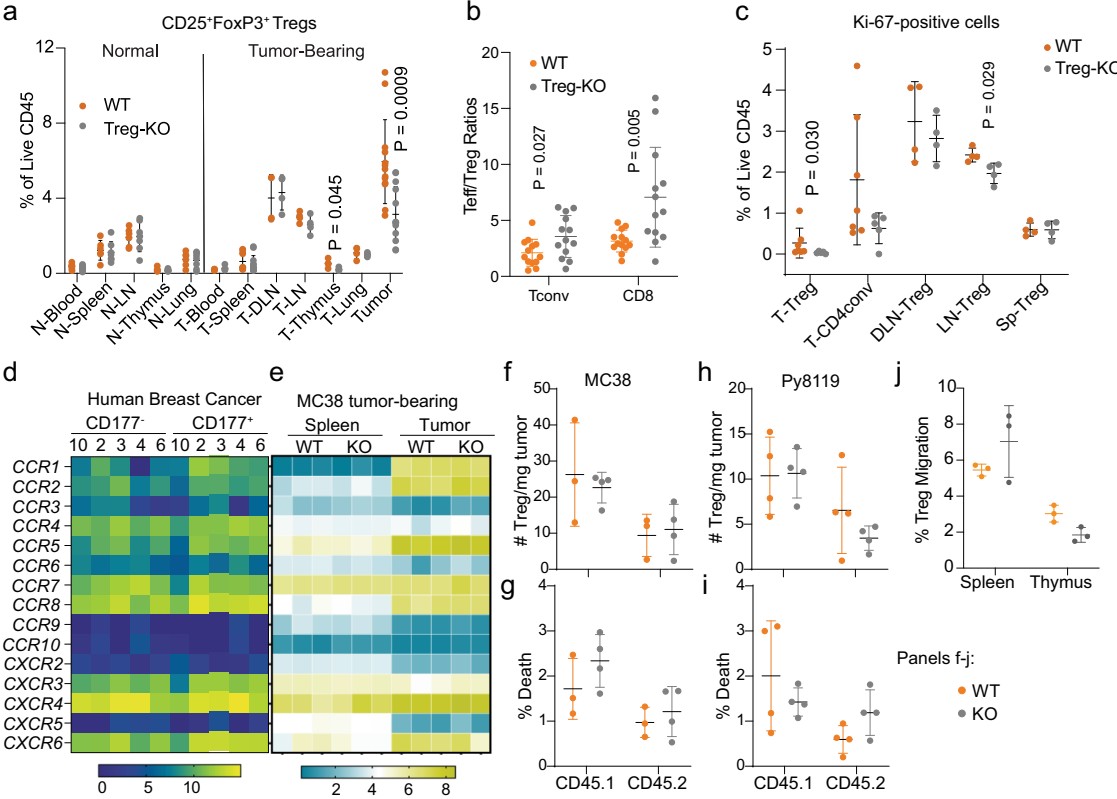

**Fig. 6 *Cd177*-deficiency in Treg cells leads to reduced frequency of TI Treg cells. a** Impact of Treg-specific *Cd177*-KO on the frequency of Treg cells within different tissues of normal or MC38-tumor-bearing mice ($n = 6$ for normal WT and $n = 7$ for KO mice; $n = 13$ for tumor TI-Treg cells and $n = 4$ for other tissues from tumor-bearing mice, T: tumor. **b** The ratios between TI Teff cells – including CD4 Tconv and CD8 T cells – and TI Treg cells in MC38 Tumors ($n = 13$ biological repeats). **c** Percentage of Ki-67+ T cells relative to total CD45+ leukocytes within different tissues from WT or Treg-specific *Cd177*-KO mice bearing MC38 tumors. Tissues were from similar experiments as in Fig. 5c tumor bearing mice ($n = 7$ WT and 5 Treg-KO). *P*-values are indicated when less than 0.05. T, tumor; DLN, draining lymph nodes; LN, non-draining lymph nodes; Sp, Spleen. **d, e** Heatmap showing gene expression of chemokine receptors within CD177+ and CD177− TI Treg cells, using flow-sorted CD177+ and CD177− TI Treg cells from 5 human breast cancer specimens (**d** data adapted from GSE89225 with patient number included), or RNA sequencing of splenic or TI Treg cells from MC-38 tumor bearing WT or Treg-specific *Cd177*-KO mice (**e** GSE150420, $n = 3$ each group; tumor KO group, $n = 2$). **f–i** Impact of CD177 on Treg recruitment into tumors. Splenic and thymic Treg cells (1:1 combined) from WT or germline CD177-KO mice of C57BL/6 background were purified (CD4+CD25+GITR+) and adoptively transferred into tumor-bearing congenic CD45.1 mice (**f–g** MC38; **h–i** Py8119). The frequency (**f, h** the number of Treg cells per mg tumor tissues) or viability (**g, i** percent death determined by Fixable Viability Dye eFluor780) of the recipient (CD45.1+) or donor (CD45.2+) TI Treg cells was determined using flow cytometry after 72 hrs of transfer ($n = 3$ WT and 4 KO tumor per group for **f, g**; $n = 4$ each group for **h, i**). **j** Impact of CD177 on Treg migration towards tumor lysates. Total splenocytes and thymocytes from littermates of WT or germline KO mice were seeded into Boyden chamber with 3 μm pore size, using 10% MC38 tumor homogenate as chemoattractant. Percent Treg migration was calculated using the number of migrated Treg cells divided by the sum of migrated and non-migrated Treg cells, determined by flow cytometry ($n = 3$ biological repeats). All data are presented mean ± s.t.d. Two-sided unpaired T-test was used for all group comparisons.

Fig. 7a–f), Treg-specific *Cd177*-KO led to a significant decrease of TI Treg cells (Fig. 6a) and some other significant changes of various immune cells within various tissues of tumor-bearing mice (Supplementary Fig. 8a–e). The ratios between TI effector T cells (CD4 Tconv or CD8) and Treg cells were significantly increased in the tumors from Treg-KO mice (Fig. 6b). We detected a decrease in thymic Treg cells caused by *Cd177* deficiency (Fig. 6a) and a decrease in proliferation as indicated by the reduced Ki-67$^+$ TI Treg cells (Fig. 6c). The expression of Treg-related genes involving Treg-suppressive function was largely unchanged between CD177$^+$ and CD177$^-$ TI Treg cells in human breast cancer (Supplementary Fig. 8f) or ccRCC (Supplementary Fig. 8g), with slight elevation of ICOS, TIGIT, and CTLA4 (Supplementary Fig. 8f, g). Chemokine receptors on TI Treg cells were not significantly different between CD177$^+$ and CD177$^-$ TI Treg cells from human breast cancer specimens (Fig. 6d), or between WT and Treg-KO TI-Treg cells from MC38 tumors (Fig. 6e). We further grew MC38 or Py8119 tumors on *Ptprc*$^a$ – a congenic CD57BL/6 J strain carrying CD45.1 – for adoptive transfer of splenic and thymic Treg cells at 1:1 ratio from non-tumor bearing WT or germline KO mice carrying the CD45.2 allele. 72 hrs after adoptive transfer, we performed flow cytometry (Supplementary Fig. 9a) and did not detect significant difference in donor TI-Treg numbers (Fig. 6f, h) or the viability (Fig. 6g, i) between WT and KO TI Treg cells infiltrating into either MC38 or Py8119 tumors. In vitro migration assay further confirmed similar migratory capability between WT and KO Treg cells towards MC38 tumor lysates (Fig. 6j). These data suggest that CD177 may not be a major regulator for the recruitment of Treg cells to tumors and that the decreased Treg number due to CD177-defeciency could be a result of decreased proliferation or thymic production of Treg cells (Fig. 6a, c).

**Expression of CD177 in TI Treg cells leads to an increased suppressive activity**. We found that CD177$^+$ or CD177$^-$ TI Treg cells, defined as CD4$^+$CD25$^+$CD127$^{low}$, from human cancer specimens exhibited a similar level of FOXP3 protein in 1 breast cancer, 2 colon cancers and 3 ccRCCs (Supplementary Fig. 9b). CD177$^+$ or CD177$^-$ TI Treg cells had similar expression of *FOXP3* mRNA in 5 breast cancers based on a published RNAseq dataset (Supplementary Fig. 9c, GSE89225). After sorting, CD177$^+$ or CD177$^-$ TI Treg cells had the same expression level of FOXP3 protein (Supplementary Fig. 9d). CD177$^+$ Treg cells had a greater suppressive activity on effector CD4$^+$ T cells (Teff) that were activated by anti-CD3/CD28 co-stimulation when plated at a 2:1 ratio compared to CD177$^-$ Treg cells (Fig. 7a). We further confirmed the superior suppression from CD177$^+$ TI Treg cells from another 5 breast cancer specimens (Fig. 7b, Supplementary Fig. 9e). Treatment with an anti-CD177-specific monoclonal antibody (MEM166) blocked the suppressive activity of CD177$^+$ TI Treg cells (Fig. 7b, Supplementary Fig. 9e), but the anti-CD177 antibody did not directly block the proliferation of CD4$^+$ or CD8$^+$ T cells without Treg cells (Supplementary Fig. 9f). The same anti-CD177 antibody is known to block CD177-mediated neutrophil transmigration[34]. We further confirmed the suppressive capacity of CD177$^+$ TI Treg cells from another independent experiment with CD177$^+$ and CD177$^-$ TI Treg cells from 7 renal cell carcinoma (RCC) specimens when Teff cells were activated by monocyte derived dendritic cells (as antigen presenting cells, APC) at 5:1 ratio (Teff/Treg) (Fig. 7c). Interestingly, CD177$^-$ Treg cells partially suppressed Teff cells when using APC (Fig. 7c), but not by anti-CD3 + anti-CD28 co-stimulation (Fig. 7a, b)—likely due to the known function of CTLA-4 from Treg cells that can complete with CD28 for CD80/CD86 ligation at the priming stage of T cell activation by APC.

Similar to the human CD177$^+$ TI Treg cells, we found that Treg cells from tumor-bearing WT mice exhibited a stronger suppressive capacity against both CD4$^+$ and CD8$^+$ T cells compared to Treg cells from tumor-bearing *Cd177*-KO mice (Supplementary Fig. 9g). CD177$^+$ TI Treg cells from mouse MC38 tumors also exhibited superior suppressive capability towards APC-activated Teff cells (Fig. 7d). Naïve CD4$^+$ or CD8$^+$ T cells purified from spleens of WT or germline *Cd177*-KO mice exhibited similar proliferative capacity upon APC stimulation (Supplementary Fig. 9h). We also purified Treg cells from tumor-bearing control or Treg-specific *Cd177*-KO mice (cKO) and confirmed the similar expression level of FoxP3 (Fig. 7e, upper left panel). TI Treg cells from control mice showed a stronger suppressive capacity against Teff cells than TI-Treg cells from cKO mice (Fig. 7e, lower left and right panels). Splenic Treg cells, however, exhibited similar suppressive capacity (Supplementary Fig. 9i).

**Discussion**

Due to the immunosuppressive role of Treg cells in peripheral tolerance, the targeting of Treg cells for cancer immunotherapy is a double-edged sword. Although the attenuation of the Treg-mediated suppressive activity increases the anti-tumor immune response, Treg cell dysfunction is often associated with auto-immune and inflammatory diseases. For example, CTLA-4 is constitutively expressed on Treg cells and acts as a primary mechanism of immune suppression by Treg cells via blocking the co-stimulation of Teff cells[35]. The anti-CTLA-4 antibody, ipilimumab, inhibits Treg function by abolishing CTLA-4-mediated suppressive signaling and depletes Treg cells in the tumor microenvironment[3,4], but the nonspecific abrogation of Treg function by ipilimumab often leads to substantial immune-related adverse effects such as colitis, which can occur in over 50% of patients. Thus, refining targetable biomarkers for TI Treg cells is critical for the development of Treg cell-targeted immunotherapy. With our expanded analysis of TI Treg cells derived from ccRCC and HCC[27], our current study represents one of the largest collections of expression profiles available for paired PB and TI Treg cells from human cancer specimens. We found a common differential signature of 143 genes comparing TI *versus* PB Treg cells. The overlapping genes included common Treg markers, such as *CTLA4*, *ICOS*, *TNFRSF4*, *TNFRSF9*, *TNFRSF18*, as well as previously identified TI-Treg markers, such as *CCR8*, *LAYN*, and *MAGEH1*. *CD177* was among the most consistently elevated genes from the single-cell RNA sequencing or the pooled RNA sequencing of TI Treg cells. CD177 is a surface protein that can be targeted by an antibody-based approach for cancer immunotherapy.

In the context of prognosis, *FOXP3* alone has had mixed results in the ability to predict overall survival for patients. More recently, two analyses of RNAseq of TI Treg cells reported the use of the ratios *CCR8/FOXP3* for breast cancer and *CCR8/CD3G* for lung and colorectal cancers, which had a modest predictive value in terms of overall survival[12,13]. Similar to algorithms developed to quantify the immune contribution of bulk RNA sequencing[36], we utilized support vector machines trained with linear kernels to develop a signature among different populations of TI Treg cells. The CF-1 signature predicted overall survival in 10 cancer types, including immunogenic cancers such as KIRC/ccRCC, SKCM, and LUAD, etc (Fig. 3). Most importantly, the CF-1 signature outperforms the newly developed common TI-Treg signature in most cancer types, as well as Treg signatures established in the literature. *CD177* gene is not included in either CF-1 or TI-Treg signatures since its expression is not Treg specific and did not pass the Treg-specific filter (Fig. 3a, Supplementary Data 4). We understand that the signature development in predicting

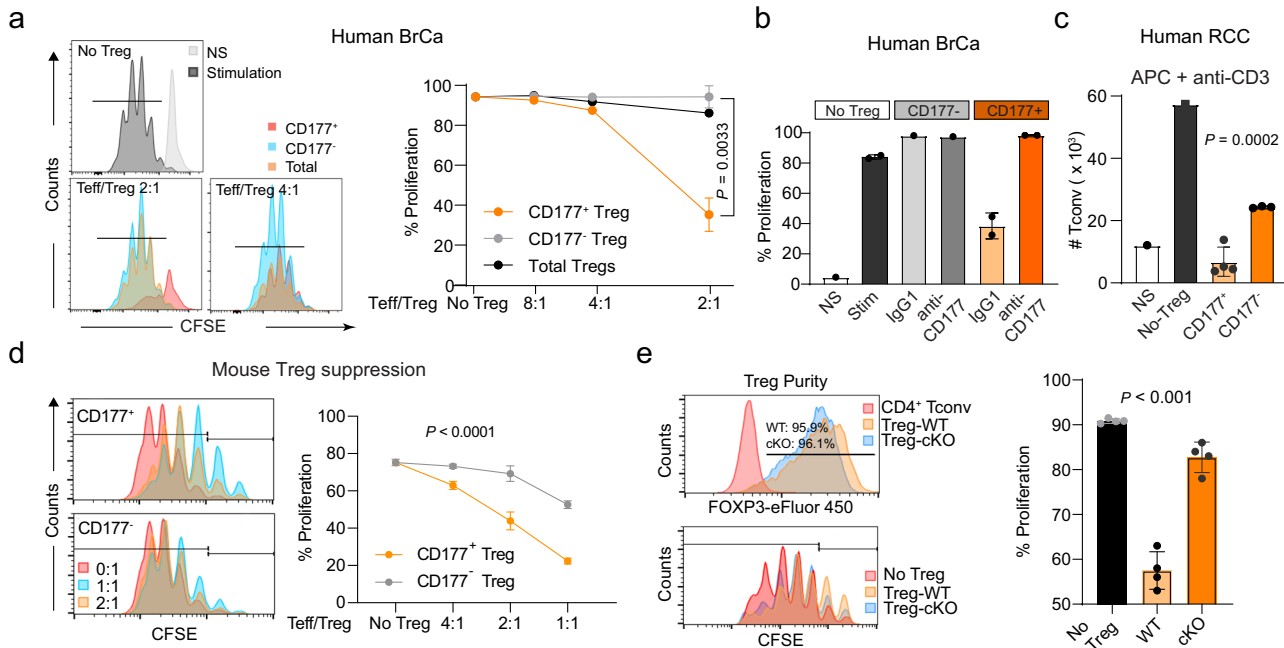

**Fig. 7 CD177+ TI Treg cells are highly suppressive to effector T cells. a** Suppressive capacity of CD177+ or CD177− TI Treg cells isolated from breast cancer specimens. CD4 Tconv cells were stimulated by anti-CD3/CD28 co-stimulation. CD4+CD25+CD127low total, CD177+ or CD177− TI Treg cells were purified from 3 individual breast cancer specimens and combined for the suppression assay. Left: Histograms showing CFSE dilution peaks indicating Teff cell proliferation and Right: Percentages of proliferative CD4 Teff cells co-cultured with total, CD177+ or CD177− TI Treg cells at ratios of 2:1, 4:1, 8:1 or no Treg (Combined data from **a** and **b**: n = 3 biological replicates for CD177+ TI Treg cells and n = 2 for CD177− TI Treg cells at 2:1 ratio; n = 1 biological replicate for other data point). P value is for comparison between CD177+ or CD177− TI Treg cells at 2:1 ratio Teff/Treg, using two-sided T-test. **b** Impact of CD177 blockade on the immune suppressive function of CD177+ TI Treg cells, using a monoclonal antibody (MEM166). Similar suppression experiments were performed as in **a**. CD177+ or CD177− TI Treg cells were purified from fresh human breast cancer specimens (combined from 5 patients) using flow cytometry and co-cultured with effector CD4 T (CD4+CD25−) cells from PBMC for ex vivo suppression assay, with or without the addition of 2 μg/ml isotype control (IgG) or anti-CD177 antibody (MEM166). Percent of proliferating cells were included and the total number of CD4 Tconv cells were enumerated after 4 days of incubation (n = 2 biological replicates for CD177+ Treg group, n = 1 for other data points). **c** Suppressive capacity of CD177+ or CD177− TI Treg cells isolated from renal cell cancer specimens (RCC). CD4 Tconv cells were stimulated by anti-CD3 and APC (monocyte derived dendritic cells). A total of 7 RCC specimens were combined (n = 1 for NS, non-stimulated or no Treg; n = 4 for CD177+ TI Treg cells; n = 3 biological replicates for CD177− TI Treg cells). The total number of CD4 Tconv cells were counted after 4 days of incubation. **d** Suppressive capacity of CD177+ or CD177− TI Treg cells isolated from mouse MC38 tumors. Effector CD4 T cells were stimulated by anti-CD3 and APC (derived from T-cell depleted splenocytes of tumor bearing mice) and combined with TI Treg cells at different ratios. Left: Histograms showing the CFSE dilution as an indicator of T cell proliferation and right: Percent proliferation as defined by the histogram (n = 3 biological replicates except n = 6 for no Treg group). Two-way ANOVA was used. **e** Impact of Treg-specific *Cd177*-KO on the suppressive capacity of TI Treg cells from MC38 tumors similar as shown in Fig. 5c. Upper left: post-sorting Treg purify was determined by intracellular staining of FoxP3; Lower left: histogram showing CFSE dilution peaks indicating CD4+ Teff cell proliferation; right: summary of percent proliferating cells (n = 4 biological replicates from 14 tumors from cKO group and 20 tumors from WT group, with 4 tumors combined in one biological replicate). All data are presented mean ± s.t.d. **c**, **e** one-way ANOVA was used.

prognosis is a rapidly evolving endeavor, but our initial validation of this CF-1-signature across TCGA cancer datasets (Fig. 3) indicates a commonality of this suppressive CF-1 signature in predicting poor prognosis. TI Treg cells have long been suspected of causing the unresponsiveness or resistance to ICIs[34,37], which was further validated in a recent study that defines the critical role of Treg cells in resistance to anti-PD-1 immunotherapy[38]. Thus, the CF-1 signature could be very informative in the context of ICI therapy, either to predict the intrinsic resistance/unresponsiveness or to predict acquired resistance to anti-PD-1 therapy.

CD177 has been mostly studied in neutrophils where it has been suggested to play a role in transendothelial migration, cell viability, and bactericidal activities. Apart from a minor point in a recent study on TI Treg cells in breast cancer where the authors found that CD177 was expressed on a small subset of TI Treg cells and a study showing that its expression is a positive prognosis marker in colorectal cancer[19], very little is known about CD177 function outside of neutrophils. We observed the presence

of CD177 in 15-25% of TI Treg cells, including breast, renal, lung, and colorectal cancers. Analysis of transcriptional heterogeneity of TI Treg cells revealed that CD177 expression was closely associated with the more suppressive CF-1 Treg cells with elevated immune suppression markers. Pseudotime trajectory places CD177 at the end stage of Treg development at the transcription level, suggesting that CD177+ Treg cells are fully differentiated and functional Treg cells, which is supported by the fact that CD177+ Treg cells are more suppressive than CD177− Treg cells in suppression assays as well as in vivo tumor models. Taken together, our results suggest that CD177 is a specific marker for a subset of suppressive TI Treg cells. Most importantly, we found that CD177 is important in mediating the immune suppressive function of TI Treg cells. We have tried many experiments to understand why CD177+ TI Treg cells are more suppressive, including (1) genetic analysis of CD177+ and CD177− TI Treg cells from mice or humans (no significant identifiable changes in the genes whose protein products are involved in immune

suppression), using either bulk RNAseq data or single cell RNA sequencing data; (2) flow cytometry for various cell surface proteins that are known to be involved in the suppressive mechanisms of Treg cells. Since both direct and indirect mechanisms of suppression have been well-established in the literature, future efforts are warranted to elucidate the molecular mechanisms with specific focus on metabolic alterations of TI Treg cells, secretion of immune-suppressive metabolites, or the identification potential CD177 receptors on effector T cells, etc.

One important question is the "targetability" of CD177 for cancer immunotherapy. As a surface protein, our data supports that blocking CD177 with antibodies can inhibit the function of CD177[+] TI Treg cells. Normal tissues have CD177 expression including neutrophils, certain epithelial cells, as well as cells within secondary lymphoid tissues. Depleting antibodies are likely to carry the risk of neutrophil depletion and the development of neutropenia. Some TI myeloid cells, likely neutrophils or granulocytic myeloid derived suppressor cells, also express CD177. It is important to know the function of CD177 in these cells. To the best of our knowledge, CD177 expression on neutrophils is constitutive and can be further increased by infections. Germline KO of *Cd177* has minimal phenotype in neutrophils even under bacterial infections, which, on one hand, indicates the safety of blocking CD177 signal; on the other hand, suggests CD177[+] TI neutrophil lineage is specific under the cancer context. Developing an anti-CD177 blocking antibody, but not a depleting antibody, has the potential to inhibit the suppressive activity of CD177[+] TI Treg cells as a complementary method for current cancer immunotherapy.

Our study highlights the importance of investigating the heterogeneity of immune cell populations, including Treg cells, in the tumors to improve the understanding of the tumor microenvironment and identify new targets for Treg-based immunotherapy. This data also provides a unique resource of transcriptome data at the single cell level from ccRCC PB and TI immune cells.

## Methods

**Patient recruitment.** The current study was approved by the University of Iowa Institutional Review Board (IRB) under the IRB number 201304826 and conducted under the Declaration of Helsinki Principles. De-identified renal cancer patients were informed and consented by Dr. Yousef Zakharia in the Department of Internal Medicine at the University of Iowa. Some human tissues were also collected from patients undergoing surgical resection after informed consent and were supplied as de-identified samples to Dr. Zhu laboratory with IRB approval (IRB number: 13-0315) or Dr. Zhang with IRB approval (IRB number: IRB201901677). Additional human specimens were collected after patient consents from the Renji Hospital, Shanghai Jiao Tong University School of Medicine in an anonymous manner to the Dr. Xuefeng Wu laboratory for cell preparation and T cell suppression assays, approved by the University Human Subject Protection Committee. All human protocols were carefully read and evaluated by the full review panels from the corresponding University Institutional Review Board (IRB), following annual review for the progress, ethics and remaining tissue procurement.

**10XGenomics library preparation, sequencing, and alignment.** The detailed single cell RNA sequencing procedures were recently published[21]. Briefly, we prepared single-cell libraries as per the 10X Genomics Chromium 5′ library and Gel Bead Kit Version 2 (10X Genomics, Pleasanton, CA). Pooled libraries were sequenced using the Illumina HiSeq 4000 in the University of Iowa Genomics Division. Basecalls were converted to FASTQ files using the Illumina bcl2fastq software and aligned to the human genome (GRCh38) using the CellRanger v2.2 pipeline. Cell quality was checked for the total expression of mitochondrial reads. Cells with <200 or >5000 unique genes were filtered out. After processing, clustering was performed using the Seurat R package (v2.3.4), correcting for patient variability using canonical correlation analysis (CCA)[49,50]. The single-cell RNA sequencing data were first normalized to correct for sequencing depth by scaling the total UMI counts per cell to 10,000. The normalized data were then log transformed after adding a pseudo count of one. Finally, each gene's log normalized counts were z-score transformed to have mean zero and a standard deviation of one. The Seurat V2 package was used to align data from the three patients by

using CCA on 1,000 highly variable genes identified using the built-in FindVariableGenes function. The top 20 dimensions of the aligned CCA were then used to cluster cells with resolution/granularity set to 1.2. Cells and clusters are visualized using t-Distributed Stochastic Neighbor Embedding (t-SNE). Differential gene expression analysis was performed using the Wilcoxon rank sum test comparing TI *versus* PB Treg cells. The p-values from the Wilcoxon rank sum test are adjusted using Bonferroni correction. For the cell trajectory analysis and pseudo-time estimates we utilized the Monocle 2 R package (v2.8.0) which is based on the reverse graph embedding algorithm[30]. The DifferentialGeneTest function in Monocle v2 was used to identify genes differing between PB and TI Treg cells. Genes with a q-value (adjusted p-value) < .01 were used to construct the trajectory. Branched expression analysis modeling (BEAM) was performed using the default settings of the Monocle 2 R package. Single-sample gene set enrichment analysis utilized the singleR R package (v0.2.0)[39] for naïve, exhausted, cytotoxicity and cell cycle gene sets that are relevant to T cells. The gene sets consisted of: (1) Cytotoxicity: *NKG7, CCL4, GZMA, PRF1, GZMA, GZMB, IFNG, CCL3*; (2) Exhaustion: *PDCD1, TIGIT, LAG3, HAVCR2, CTLA4*; (3) naïve *CCR7, TCF7, LEF1, SELL*; (4) Effector/ memory: *CD27, CD28, CCR7, CCR5, SELL* and *FAS*. Cell cycle regression for individual Treg cells was performed with the Seurat p Package, as previously described[40]. For ccRCC data, the quantified gene expression counts and V(D)J T cell receptor sequences for single-cell RNA sequencing are available at the Gene Expression Omnibus (GEO) at GSE121638. The same processing and quality control procedures were applied to count-level data from the HCC single cell RNA sequencing dataset: GSE98638 as well[27].

**Bulk RNA sequencing Treg data.** Raw expression data for GSE89225[12] and PRJEB11844[13] were downloaded from the NCBI Sequence Read Archive and the European Nucleotide Archive, respectively. SRA files were converted to FASTQ files using the SRA toolkit. Samples were aligned with the kallisto pseudoalignment protocol and GRCh38 build for the human genome to produce estimated counts[41]. Treg expression values were processed using the Sleuth R Package (v0.30.0)[42].

**Statistical analysis.** Statistical analyses were performed in R (v3.5.1). Two-sample significance testing utilized Welch's T test, with significance testing for more than three samples utilizing one-way analysis of variance (ANOVA) with Tukey honest significance determination for correcting multiple comparisons. Differential gene expression for single-cell RNA-seq data was performed in the Seurat R package using the Wilcoxon rank sum test with *p*-values adjusted using the Bonferroni method. Differential gene expression of bulk RNA-sequencing Treg expression data utilized Wald testing in the Sleuth R Package[42]. Differential gene expression between branches of the cell trajectory was performed in the Monocle R package using likelihood ratio testing[30]. All other statistics were performed using embedded functions of GraphPad Prism (v8.3.0).

**Signature development.** Log2 expression values from the Cancer Genome Atlas (TCGA) were acquired from the UCSC Xena Browser. Updated survival information was obtained from the recent work of Liu, et al[43]. After merging expression and clinical outcome data, an exhaustive best subset selection via the branch-and-bound algorithm in the leaps library (v3.0) in R was performed to identify features most predictive of overall survival. Feature selection was performed for three sets of genes: 1) 143 differentially-expressed genes shared between ccRCC and HCC TI Treg cells, 2) 86 genes differentially expressed in CF2 and 3) 222 genes differentially expressed in CF1 using the KIRC TCGA cohort. These genes were then filtered for TI-Treg-specific genes based on the comparisons with other tumor-infiltrating immune cell lineages in the ccRCC data set. A total of 226 genes were found to be significantly upregulated with log-fold change ≥ 1 and FDR < 0.05 in at least 3 of the 5 immune lineage comparisons.

To ensure the computational feasibility of best subset selection, a custom scoring algorithm was devised. In each of the three gene sets, a random group of <30 genes was used for best subset selection, and the five genes that compromised the best five-predictor model were given a point. This process was repeated with a new random group of <30 genes until all genes had been included in exactly 500 best subset selection models. The frequency of inclusion in the top-performing five-predictor model was assessed for each gene. The six genes that were most frequently included in the top performing five-predictor model were utilized for training subsequent machine learning algorithms. For each set of feature-selected genes, support vector machines (SVMs) were trained to predict overall survival using the e1071 (v1.7.0) R package. The 533 patients in the TCGA KIRC dataset were randomly divided into training (10%, n = 53) and testing (90%, n = 480) cohorts. SVMs were trained to discriminate overall survival status with linear kernels, and the cost parameter was selected via cross-validation. Kaplan-Meyer curves were constructed with the survival (v2.42.6) and survminer (v0.4.3) R packages. The Cox proportional hazards regression model within the survival package was used to compute the hazard ratios between good-outcome and poor-outcome prediction groups.

**Cell lines and cell culture.** PY8119 cells were derived from a primary *MMTV-PyMT* tumor in the C57BL/6 background as described previously[44] and were

provided by Dr. Lesley Ellies at the University of California, San Diego before she provided to ATCC (CRL-3278). The cells were maintained in F12 media supplemented with 10% fetal calf serum (FCS), 10 ng ml$^{-1}$ epithelial growth factor, 2 μg ml$^{-1}$ hydrocortisone, and 5 μg ml$^{-1}$ insulin. Cells were detached using 2.5% trypsin and resuspended in 1:1 phosphate buffer saline: Matrigel (Corning, Corning, NY) mixture and injected in mammary fatpad at 100 μl volume per inoculation. MC38 colorectal cells were provided by Dr. Yuwen Zhu laboratory at the University of Colorado Denver and were maintained in the same complete F12 media, detached using 2.5% trypsin, and resuspended in phosphate buffer saline and injected into mammary fatpad at 100 μl volume per inoculation. The cell lines were tested mycoplasma negative using Mouse Essential CLEAR Panel by Charles River Research Animal Diagnostic Services (CR CADS, Wilmington, MA).

**Flow cytometry.** Mouse tumors and tissues were excised and digested using the mouse Tumor Dissociation Kit (Miltenyi Biotec) to obtain single cell suspensions. Red blood cells were lysed using ACK lysis buffer. Mouse cells were then washed and incubated with combinations of the following antibodies: anti-mouse CD62L-BV785 (clone MEL-14), anti-mouse MHCII I-A/I-E-BB515 (Clone 2G9, BD Biosciences), anti-mouse CD11B-PEdazzle (clone M1/70), anti-mouse CD45-AF532 (clone 30 F.11), anti-mouse CD3-APC (clone 17A2), anti-mouse CD8-BV510 (clone 53-6.7), anti-mouse CD4-BV605 (clone GK1.5), anti-mouse CD11C-PE-Cy7 (clone N418), anti-mouse Ly6G-FITC (clone IA8), anti-mouse Ly6C-BV711 (clone HK1.4) anti-mouse F4/80-BV650 (clone BM8), anti-mouse CD80-BV480 (clone 16-10A1, BD Biosciences), anti-mouse CD25-PE-Cy5 (clone PC61) plus FVD-eFluor-780 (eBioscience), anti-CD19 (6D5), anti-B220-PerCP0EF710 (RA3-6B2, eBiosciences) and mouse FcR blocker (anti-mouse CD16/CD32, clone 2.4G2, BD Biosciences). After surface staining, cells were fixed and permeabilized using the FOXP3/Transcription Factor Staining Buffer Set (eBioscience). Cells were stained with a combination of the following antibodies: anti-mouse FOXP3-EF450 (clone FJK-16S, eBioscience), anti-mouse Ki-67-PerCP-Cy5.5 (clone 16 A8). Most antibodies are from Biolegend unless indicated otherwise, with 1:100 dilution per recommendations unless specifically indicated in Supplementary Table 1, including antigen name, clone number, company information, catalog numbers, fluorophores, and dilutions.

**Animals.** All animals were maintained under specific pathogen-free conditions according to the IACUC guidelines. *Cd177*-knockout mouse model was constructed in our previous publication[17]. The Treg KO of *Cd177* is confirmed by using real-time PCR with primers listed (Supplementary Fig. 6A and Table 2). Mice carrying *Cd177*$^{fl/fl}$ were developed by the Genome Editing Facility, University of Iowa using CRISPR/Cas9 technology. The targeting strategy scheme is in the Supplementary Fig. 5. Mice were genotyped using primers targeting both 5' and 3' LoxP sites. 5' LoxP site primers are: 5'-GTGTTGCGTTTCCTGCCTTG; 5'-CTGGTTACCTTATGCCACTCC with a 172 bp wildtype product and 210 bp mutant. 3' LoxP site primers are: 5'-GGGTTGCCAAGACTTGATAATG; 5'-AGGTGAGACACTAGAGAAGAGG with a wildtype 164 bp product and mutant 203 bp product following standard PCR protocol using DNA extracted from mouse tail snips (Supplementary Table 2). The targeting allele was sequenced to confirm the correct insertion of LoxP site at both genomic locations. Treg-specific *Cd177*-KO was achieved by mating *Cd177*$^{fl/fl}$ mice with *Foxp3*$^{YFP/Cre}$ (Stock No.: 016959, Jackson Laboratories, Bar Harbor, Maine). Male mice carrying *Cd177*$^{fl/fl}$ and hemizygous *Foxp3*$^{YFP/Cre}$ or female mice carrying *Cd177*$^{fl/fl}$ and homozygous *Foxp3*$^{YFP/Cre}$ were used for experimental groups. We used 7-8 week old mice for tumor study and euthanized the mice between 12-16 weeks of age. All animal studies were approved by the University of Florida Institutional Animal Care and Use Committee (IACUC) under protocol 201810399. All animals were housed in the same room of a specific pathogen-free (SPF) within the Association for Assessment and Accreditation of Laboratory Animal Care accredited facility at the University of Florida and performed in accordance with IACUC guidelines. Euthanasia was performed by the inhalation of medical grade CO2 per current AVMA recommendations and UF IACUC policies. The WT and germline CD177-KO mice were housed separately. The control group and Treg-specific *Cd177*-KO group were co-housed. The room temperature is 21.1–23.3 °C with an acceptable daily fluctuation of 2 °C. Typically the room is 22.2 °C all the time. The humidity set point is 50% but can vary ±15% daily depending on the weather. The photoperiod is 12:12 and the light intensity range is 15.6–25.8 FC.

**Adoptive transfer of Treg cells.** MC38 or PY8119 tumors were similarly grown in *Ptprc*$^a$ (CD45.1) mice (Jackson Laboratory) to 1 cm in diameter. EasySep Mouse CD4$^+$CD25$^+$ Regulatory T Cell Isolation Kit (StemCell) was used to enrich splenic and thymic CD4$^+$CD25$^+$ cells from WT and germline CD177-KO C57BL/6 J (CD45.2$^+$) mice. A total of 3 ×10$^5$ CD177 Treg cells from WT or KO mice (1:1 ratio of splenic and thymic Treg cells) were resuspended in 50 ul PBS for the adoptive transfer into the tumor-bearing CD45.1$^+$ *Ptprc*$^a$ (CD45.1) mice via retroorbital route. 48 hrs later, mice were euthanized, and tumors were collected to quantitate either recipient (CD45.1$^+$) or donor (CD45.2$^+$) TI Treg cells.

**Tissue collection and T cell suppression assay.** Fresh human breast cancer and renal cancer samples were collected from patients undergoing surgical resection after informed consent, detailed above. Breast cancer samples were supplied de-identified by the Tissue Procurement Core at the University of Iowa Hospitals and Clinics and were histologically characterized by the Department of Pathology at the University of Iowa according to IHC of ER, PR and HER2 biomarkers. De-identified blood samples from healthy donors were provided by the blood bank at the University of Iowa Hospitals and Clinics. For renal cancer patients, de-identified matching blood samples were provided by the Genito-Urologic Molecular Epidemiology Resource at the Holden Comprehensive Cancer Center, University of Iowa Hospitals and Clinics. Human PB mononuclear cells (PBMCs) were isolated from whole blood using Ficoll Plaque (GE Healthcare Biosciences, Pittsburgh, PA) density gradient centrifugation or SepMate Tubes (StemCell Technologies).

Fresh tissues were directly distributed to the research laboratories after surgery, followed by enzymatic digestion and physical dissociation using gentleMACS (Miltenyi, Bergisch Gladbach, Germany) as per manufacturer's instruction. Cell suspensions were filtered through a 100 μM cell strainer, magnetically enriched using anti-CD45 positive selection (Miltenyi) for TI leukocytes, and immediately frozen using FBS with 5% DMSO.

For in vitro suppression assay, naive CD4$^+$ T cells were isolated from PBMC using the human naïve CD4$^+$ T cell isolation kit II (Miltenyi, 130-094-131). Naïve CD4$^+$ T cells were stimulated by anti-CD3/28 dynabeads (Thermo Fisher) and 10 ng/ml IL-2 (R & D) overnight for co-culturing with Treg cells for Fig. 7a, b; or with monocyte-derived dendritic cells (Fig. 7c) from human PBMC with IL-4 (500U/ml, Immunotools, Friesoythe, Germany) and GM-SCF (800 U/ML, Immunotools). TI-Treg cells were labeled with anti-CD45, anti-CD4, anti-CD25, anti-CD127 antibodies and flow-sorted. We combined Treg cells from 3 human breast cancer specimens to get enough number of cells for Fig. 7a, 5 breast cancer specimens for Fig. 7b, 7 RCC specimens for Fig. 7c, CD45$^+$CD4$^+$CD25$^+$CD127$^{low}$ Treg cells were used as total TI Treg cells, in addition to further separation of CD177$^+$ or CD177$^-$ TI Treg cells. CFSE-labeled pre-activated naïve CD4$^+$ T cells were co-cultured with purified TI-Treg cells at the indicated ratios (2:1, 4:1, or 8:1) on 96-well round-bottom plates – maintaining continuous stimulation by anti-CD3/28 dynabeads (Thermo Fisher) or monocyte-derived dendritic cells in RPMI1640 supplemented with 10% fetal bovine serum (FBS), 10 mM HEPES, 2×10$^{-5}$ M 2-mercaptoethanol for 96 hr. For antibody blocking assay, we used either isotype control IgG or anti-CD177 (clone MEM166, Biolegend) at 2 μg/mL concentration with the 2:1 Teff/Treg ratio for CD3/28 dynabead stimulation and 5:1 Teff/Treg ratio for monocyte-derived dendritic cell stimulation.

**Antibody staining and multicolor immunophenotyping of TI-Treg cells.** Multicolor phenotypic panels were established using different combinations of fluorescently tagged anti-CD45 (H130), CD3 (HIT3a), CD4 (OKT4), CD25 (M-A251), CD27 (O323), CD127 (A019D5), CCR8 (433H), PD-1 (EH12.2H7), CTLA-4 (BN13), FOXP3 (206D), and CD177 (MEM-166). Cells were stained using standard immunofluorescent staining protocol and run on flow cytometry either using live cells or fixed in 4% paraformaldehyde. Hoechst or eBiosciences Fixable Viability Dye eFluor 780 (ThermoFisher, eBiosciences) was used to exclude death cells. Antibodies were purchased from Biolegend, Molecular Probes, BD Biosciences and eBiosciences. Flow cytometric data were acquired on a 4-laser LSR II (BD Biosciences) or Cytek Aurora (Cytek, Fremont, CA) and data were analyzed using FlowJo software (TreeStar, Ashland, OR). For experiments using frozen samples, cells were thawed and suspended in RPMI supplemented with 10% FBS and incubated for 1.5 h at 37°C, 5% CO$_2$ before staining. Treg cells were identified as CD3$^+$CD4$^+$FOXP3$^+$ or CD3$^+$CD4$^+$CD25$^+$CD127$^{low}$. All human antibodies are listed under Supplementary Table 1, including antigen name, clone number, company information, catalog numbers, fluorophores, and dilutions.

**In vitro transmigration assay.** Total splenocytes and thymocytes were isolated from 6 to 8 weeks of male C57BL/6 J mice. Isolated cells were resuspended in 0.1% FBS PBS, with 5 ×10$^6$ cells plated into the top chamber of the 24-well plate transwell system (Corning, 3 μm size pore). A 10% MC38 tumor homogenate was prepared in RPMI was added to the lower chamber as a chemoattractant. 24 hrs later, cells from top chambers and migrated towards tumor homogenates were collected for flow cytometry to enumerate percent Treg migration, by using migrated Treg cells divided by the total number of Treg cells (top and migrated Treg cells).

**Immunohistochemistry (IHC) staining.** For IHC staining, formalin-fixed, paraffin-embedded tissue samples were sectioned at 5 μm by the University of Iowa Comparative Pathology Laboratory. Sections were then de-paraffinized in a series of xylene washes and rehydrated with ethanol and water. Antigen retrieval was accomplished by immersing slides in a citrate buffer (pH 6) solution in a 110 °C water bath for 15 min. Slides were allowed to cool-to-room temperature and were then incubated with 3% hydrogen peroxide followed by 2–5 min washes in 1X Dako buffer. Slides were then incubated with mouse monoclonal anti-CD177 antibody (clone C4C; Sigma) for 1 h and then Dako Mouse Envision System HRP for 30 min. The slides were developed with 3,3'-diaminobenzidine (Dako DAB

plus) for 5 min followed by 3-min incubation in 3,3′-diaminobenzidine tetra-hydrochloride (Dako DAB enhancer). Samples were then counterstained with hematoxylin, rinsed in water, cleared in xylene and then mounted with Permount. Histological evaluations were performed by comparative pathologist Dr. Katherine Gibson-Corley.

**Reporting summary**. Further information on research design is available in the Nature Research Reporting Summary linked to this article.

## Data availability

All original data are available upon request. We included new analysis from publicly available datasets including: GSE89225, PRJEB11844, and GSE98638. The single cell RNA sequencing result for renal cancer is deposited as GSE121638 and the TI-Treg cells from the MC38 tumor model is deposited as GSE150420. TCGA datasets were downloaded for the TCGA website or UCSC Xena Browser (https://xena.ucsc.edu/) including KIRC/ccRCC, breast cancer, melanoma and 21 other cancer RNA sequencing datasets and clinical parameters. Source data are provided with this paper.

## Code availability

All codes related to the single cell RNA sequencing is available here: https://github.com/ncborcherding/.

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

## Acknowledgements

We thank the Breast Molecular Epidemiologic Resource (BMER led by Dr. Sonia Sugg) and Tissue Procurement Core at the University of Iowa Carver College of Medicine for providing breast cancer tissues; TCGA for providing breast cancer, melanoma and other cancer RNA sequencing datasets and clinical parameters; Comparative Pathology Laboratory from Department of Pathology University of Iowa Carver College of Medicine for developing CD177 IHC protocol. The work was partially supported by NIH

grants CA200673 (W.Z.), CA203834 (W.Z.), CA260239(W.Z.), AI067846 (D.A.), CA29655 (N.B.). DOD/CDMRP grant BC180227 (W.Z.). Y.Z. was supported by funding from the Rock 'n' Ride. W.Z., D.A., and R.B. were also supported by grants from the UF Health Cancer Center. This work is also supported by the University of Iowa Holden Comprehensive Cancer Center support grant P30 CA086862.

## Author contributions

Conception and design: W.Z., N.B. K.K.A, X.W., Y.Z. Development of methodology: N.B., K.K.A., P.K., K.G.C., A.P.V., X.Z., U.D., T.D., E.H. Acquisition of cancer specimens and data: K.K.A., N.B., R.K., A.P.V., A.V., P.K., G.P., H.X., J.H., Zheng Wang, H.L., D.W., Zhengting Wang, J.C., M.L., H.Z., J.T., M.K., K.G.C., J.K.T., Y.W.Z., J.L., X.H., X.W., W.Z. Bioinformatics: N.B., A.P.V., R.B. Analysis and interpretation of data: N.B., K.K.A., R.K., X.W., Y.W.Z., W.Z. Writing, review, and/or revision of the manuscript: N.B., K.K.A R.K., R.B., Y.W.Z, S.G.Z, D.A., D.Z., Y.Z., W.Z. Study supervision: X.W., Y.Z., W.Z.

## Competing interests

X.Z. and D.Z. are inventors of two pending patent applications for the use of BCL-$X_L$ PROTACs as senolytic and antitumor agents. D.Z. is the co-founder of, and D.Z. and W.Z. have equity in, Dialectic Therapeutics, which develops BCL-$X_L$ PROTACs for the treatment of cancer. The other authors have no competing interests.

## Additional information

[1]Department of Pathology, Immunology and Laboratory Medicine, University of Florida, Gainesville, FL 32610, USA. [2]UF Health Cancer Center, University of Florida, Gainesville, FL 32610, USA. [3]Department of Pathology, University of Iowa, Iowa City, IA 52242, USA. [4]Cancer Biology Graduate Program, University of Iowa, Iowa City, IA 52242, USA. [5]Medical Scientist Training Program, University of Iowa, Iowa City, IA 52242, USA. [6]Department of Pharmaceutics and Translational Therapeutics, College of Pharmacy, University of Iowa, Iowa City, IA IA52242, USA. [7]Department of Anatomy and Cell Biology, University of Florida College of Medicine, 32610, Gainesville, FL 32610, USA. [8]Department of Pharmacodynamics, College of Pharmacy, University of Florida, Gainesville, Fl 32610, USA. [9]Department of Pathology, Microbiology & Immunology, Vanderbilt University Medical Center, Nashville, TN 37232-2130, USA. [10]Department of Internal Medicine, University of Iowa, Iowa City, IA 52242, USA. [11]Department of Surgery, University of Colorado Anschutz Medical Campus, Aurora, CO 80045, USA. [12]Department of Breast Surgery, Renji Hospital, Shanghai Jiao Tong University School of Medicine, Shanghai, Shanghai 200127, China. [13]Shanghai Institute of Immunology, Department of Immunology and Microbiology, Shanghai Jiao Tong University School of Medicine, Shanghai 200025, China. [14]Shanghai Key Laboratory for Tumor Microenvironment and Inflammation, Key Laboratory of Cell Differentiation and Apoptosis of Chinese Ministry of Education, Department of Biochemistry and Molecular Cell Biology, Shanghai Jiao Tong University School of Medicine, Shanghai 200025, China. [15]State Key Laboratory of Oncogenes and Related Genes, Shanghai Jiao Tong University School of Medicine, Shanghai 200025, China. [16]Department of Gastrointestinal Surgery, Renji Hospital, School of Medicine, Shanghai Jiao Tong University, Shanghai 200025, China. [17]Department of Nuclear Medicine, Ruijin Hospital, Shanghai Jiao Tong University School of Medicine, Shanghai 200025, China. [18]Department of Gastroenterology, Ruijin Hospital, Shanghai Jiao Tong University School of Medicine, Shanghai 200025, China. [19]Department of Medicine, Division of Rheumatology & Clinical Immunology, University of Florida, 1600 Archer Road, Gainesville, FL 32610-0275, USA. [20]Department of Immunology, Moffitt Cancer Center, Tampa, FL 33612, USA. [21]Department of Biostatistics, University of Florida, Gainesville, FL 32610, USA. [22]Department of Internal Medicine, Ohio State University College of Medicine and Wexner Medical Center, Columbus, OH 43210, USA. [23]Hongqiao International Institute of Medicine, Shanghai Tongren Hospital, Shanghai Jiao Tong University School of Medicine, Shanghai 200025, China. [24]Present address: College of Pharmacy, University of Baghdad, Department of Pharmaceutics, Baghdad 10071, Iraq. [25]Present address: Department of Radiation Oncology, Washington University School of Medicine, St. Louis, MO 63110, USA. [26]These authors contributed equally: Myung-Chul Kim, Nicholas Borcherding, Kawther K. Ahmed, Andrew P. Voigt. ✉email: xuefengwu@shsmu.edu.cn; yousef-zakharia@uiowa.edu; zhangw@ufl.edu

