## [Peer Review File · Nature Communications]

Reviewers' comments:

Reviewer #1 (Treg, TCR signalling, general T biology)(Remarks to the Author):

This paper can be divided into two parts. The first describes a unique transcriptional signature of tumor infiltrating Tregs. The second part of the paper describes the function of CD177, a neutrophil antigen, which the investigators identified as a component of the Treg signature expressed on the TI Tregs. Unfortunately, both parts of the paper suffer from major methodology problems:

1. The authors use the TI signature found in the CF1 component of Treg gene expression to predict survival. One of the major genes found in CF1 is CD177. However, in the Discussion of the paper on line 216 the authors state the CD177 is not included in either CF1 or the TI-Treg signature due to lack of specificity. This fact is never mentioned in the results which surely give the impression that the expression of CD177 is a major predictor of a poor prognosis. In any case, the entire interpretation of the significance of the CF1 component has to be revised. This is very misleading.

2. The second part of the paper describes a potential role for CD177 in Treg suppressor function. Approximately 20% of TI-Treg are said to express CD177, an antigen which has never been described to be expressed on T cells. Most of this data is derived by gating on CD225+CD127⁻ Tregs, not on Foxp3⁺ T cells (Fig.4). The problem with this approach is that the expression of CD25+CD127^{lo} may not be bona fide markers of Treg derived from tumors. The authors must show that the purity of their populations and gate on expression of Foxp3 or perhaps Helios which are the only Treg markers that should be used. In fact, in Sup Fig 4, where the authors actually gate on Foxp3 a very high percentage of Foxp3⁺ T cells express CD177. However, this appears to be a single tumor sample.

3. The characterization of expression of CD177 needs to be significantly expanded by studying both resting Treg, activated Treg, resting T conventional and activated T conventional cells at the cell surface and mRNA level. Could some of the low levels of staining be secondary to doublets between T cells and neutrophils.

4. The Treg suppression assays have not been performed correctly. One NEVER adds IL-2 to a Treg suppression assay as studies performed by the Sakaguchi/Shevach labs more than 2 decades ago clearly demonstrated that the addition of IL-2 abrogates or masks the suppressor function of Tregs. Again all the populations used in Fig. 5c must be stained for Foxp3 expression. The authors characterize these populations in the figure legend as Foxp3⁺, but never show this data and expression of CD25 and Cd127 as pointed out above are not equivalent. Why don't the total Treg population suppress? Are the Treg populations used non-responsive to either TCR or IL-2 stimulation as is characteristic of Foxp3⁺ Treg in both mouse and man?

5. The reversal of suppression by the anti-CD177 is a very intriguing observation as it implies that CD177 is actually involved in the in vitro function of Tregs. This claim requires much greater experimentation. First, the Treg suppression assay needs to be performed in the absence of IL-2 using standard conditions. Secondly, CD177 is a member of the Ly-6 family of antigens and antibodies to this family of antigens in either mouse or man (e.g, CD59, JI 146, 4092, 1991) have potent T cell activating properties under certain conditions in vitro. The authors must prove that the Treg is the actual target for the effects of the mAb and not the responding T cell. What cells are actually dividing in Fig. 5D?

6. The studies using global and Treg conditional knock out mice are incomplete. The phenotype of both of these mice must be completely characterized. It remains unclear to me if any mouse T cell expresses CD177. Sup Fig 7B is described in the text at showing mouse splenocytes yet the figure and the legend indicate human splenocytes. There are potentially numerous explanations why CD177 deficient Treg fail to suppress in the mouse tumor model. For example, one study has suggested that PECAM-1 is actually the ligand for CD177 and it is therefore possible that the CD177 deficient Tregs fail to enter the tumor microenvironment and thus fail to enhance tumor growth. A complete work up of Treg function in these mice is required before any conclusions can be drawn from the data presented. Commercial mAbs to mouse CD177 are available. It is not clear what mAb is used to

identify mouse CD177 in this study.

Reviewer #2 (Treg, TCR signaling)(Remarks to the Author):

Here, the authors analyze gene expression signatures of individual Treg isolated from human tumors and peripheral blood. They identify sets of up-regulated and down-regulated genes in comparative analysis of TIL and PBL Treg, consistent with previously published data sets of tumors of different origin. The finding that CD177 identifies a particularly suppressive subset of Treg is interesting, as are the results of knocking out CD177, either constitutively or inducibly. However, it is not clear from the manuscript why the authors chose to focus on CD177 (other than stating later that it was upregulated in a subset of Cluster 1). It is also not clear from the manuscript how the authors think CD177 may be acting to modify Treg function (i.e. apparently increasing their suppressive activity).

Regarding the single cell RNAseq analysis, the authors do not report how many Treg cells are being analyzed in each case, or in total (what is reported is number of immune cells and not the number of Treg). This should be clarified.

In Fig. 2, the authors, based on pseudotime analysis, discuss the origination of two clusters of Treg. However, they do not mention that some junctional zone and liver Treg have a similar expression profile (Supplementary Fig. 2A). Thus, the authors should address the fact that these variables may be a common of Treg in non-immune tissues in general, and may not be specific to tumors.

In Fig. 2 D-E and Supplementary Fig. D-E, the authors imply that cluster 1 and 2 have an exhaustion signature. However, it is not clear which genes are being used to define these cells, nor what the cytotoxicity, cell cycle and naïve status signature scores are based on. Of note, "exhaustion" of Treg is not well-defined, particularly at a functional level.

In Fig. 4A, the authors should clarify whether the tissue-infiltrating cells from liver show any expression of CD177; also, the status of CD177 in tumor-associated tissue used in the previous figure is not mentioned.

In 4C, the authors label CD127 on both the X and Y axes. I think the Y axis should probably be CD177.

In Supplementary Fig. 4D, where the authors show staining and gating controls for Treg analysis, there seems to be a nonspecific shift of the whole CD3 population, compared to the FMO control. Moreover, the sequence of flow analysis in Supp. Fig 4D is odd, where FoxP3 has been used to gate before gating on CD4. It would be better to perform the gating the other way around, i.e. first gate on CD4, then show FoxP3, etc.

Fig. 5 and the supporting supplementary figure state are meant to show that CD177 positive Treg do show upregulation of other markers, like PD-1 and CTLA-4. Have the authors looked to see if other molecules associated with potent Treg suppression, like some cytokines and granzyme B, are upregulated in their staining?

In Fig. 6, the authors use mice with global or Treg-specific KO of CD177. However, they do repeat the in vitro suppression assay with TIL Treg from either one of these mouse models. In addition, they do not perform any lymphocyte characterization (including of Treg), before proceeding to the tumor models. The authors need to address directly whether the global or inducible KO of CD177 leads to any baseline changes in lymphoid organs. Also, the authors fail to disclose which FoxP3-Cre they are using in their Treg CD177 KO mouse. In addition, in the tumor experiments there is no detailed TIL

characterization to say whether total or Treg-specific KO of CD177 leads to any change in various lymphocyte compartment inside the tumors.

Finally, since the authors have access to a CD177 blocking antibody, they could test it in the tumor models with WT mice, although I would consider this experiment a lower priority in the context of other issues that should be addressed, as discussed above.

Reviewer #1 (Treg, TCR signalling, general T biology) (Remarks to the Author):

1. The authors use the TI signature found in the CF1 component of Treg gene expression to predict survival. One of the major genes found in CF1 is CD177. However, in the Discussion of the paper on line 216 the authors state the CD177 is not included in either CF1 or the TI-Treg signature due to lack of specificity. This fact is never mentioned in the results which surely give the impression that the expression of CD177 is a major predictor of a poor prognosis. In any case, the entire interpretation of the significance of the CF1 component has to be revised. This is very misleading.

Response: Thanks for the comment. We understand the exclusion of CD177 is confusing, but based on our recent publication (1), CD177 is expressed by epithelial cells in normal breast and other tissues (New Supplementary Figures 4D-E showing IHC staining for CD177 in various tissues). Most importantly, the cancer/epithelial cell-intrinsic CD177 is dominant if expressed and is associated with a better prognosis in colon (2), gastric (3) and breast cancers (1), in contrast to its role in TI-Tregs where CD177 should promote cancer immunity and be correlated with bad prognosis. This will complicate the analysis using CD177 as a TI-Treg signature. We added the citation in the introduction as well. In addition, the signature was generated using the computer-learning algorithm. Genes enriched in CF1 were further selected by screening Treg-specific gene list (Supplementary Table 3, generated by comparing DEGs between TI-Tregs and all other immune cells in the ccRCC single cell RNAseq datasets, highlighted). This prevents the bias of the signature selection process.

2. The second part of the paper describes a potential role for CD177 in Treg suppressor function. Approximately 20% of TI-Treg are said to express CD177, an antigen which has never been described to be expressed on T cells. Most of this data is derived by gating on CD25+CD127- Tregs, not on Foxp3+ T cells (Fig.4). The problem with this approach is that the expression of CD25+CD127lo may not be bona fide markers of Treg derived from tumors. The authors must show that the purity of their populations and gate on expression of Foxp3 or perhaps Helios which are the only Treg markers that should be used. In fact, in Sup Fig 4, where the authors actually gate on Foxp3 a very high percentage of Foxp3+ T cells express CD177. However, this appears to be a single tumor sample.

Response: The antigen CD177 expression on Tregs has been reported by Rudensky laboratory, but they didn't include any functional elaboration – only based on RNAseq of pooled CD177+ and CD177- Tregs from breast cancer specimens (4, 5). In addition, we now include the co-staining of Foxp3, CD25 for better defining the Treg populations from 5 human breast cancer specimens (we cannot get more fresh tissues due to COVID-19 influence on our clinic). Representative data is updated in Figure 4D and Supplementary Figure 4G with updated gating as suggested. Similarly as shown in the new Supplementary Figure 4H (original Figure 4C), all specimens have a certain amount of CD177+ Tregs defined by CD4+CD25+FoxP3+, but Tconv or CD8 T cells have no or little CD177+ cells.

3. The characterization of expression of CD177 needs to be significantly expanded by studying both resting Treg, activated Treg, resting T conventional and activated T conventional cells at the cell surface and mRNA level. Could some of the low levels of staining be secondary to doublets between T cells and neutrophils.

Response: We re-analyzed CD177 expression on ccRCC single cell RNAseq data and only TI-Treg has CD177 expression and no other population exhibit CD177 mRNA, including active/resting CD4 conventional and different CD8 populations (New Figure 4C). The single cell RNAseq data is very similar as the protein data (New Figure 4D, supplementary Figure 4G). The expression of CD177 on tumor-infiltrating T cells is nearly exclusive to TI-Tregs.

For Treg activation/resting states, we cannot continue to collect fresh cancer specimens for flow cytometry due to COVID-19, but we re-analyzed some of old flow data and found that CD177⁺ Tregs are exclusively CD45RA⁻ (Figure attached below, from a breast cancer patient), indicative of effector phenotype, in agreement with known literature that cancer-associated Tregs are mostly effector/near-maximal active Tregs (4, 6, 7), including breast cancers (8).

It is very clear that resting or activation states of Tregs are not the determinant of CD177 expression because human PBMC Tregs contain both resting and active Tregs and there is no CD177 expression on PBMC Tregs at all. Among CD4⁺ Tconv cells or CD8⁺ T cells, activation or resting state is not relevant to CD177 expression since there hardly any CD177 positivity on Tconv or CD8⁺ T cells (averaged 0.31% Tconv cells express CD177 in breast cancer and 0.72% in renal cancer (Figure 4E), similarly as mRNA level above (Figure 4C).

Related to doublet: Please note all our gating is based on lymphocytes area when analyzing Tregs as shown in Figure 4D and supplementary Figure 4H (Lym gating), using very stringent lymphocyte/single cell gating using blood or spleen as guide, at the same time removing doublet by FSC-A versus FSC-H; SSC-A versus SSC-H. In addition, we always used CD8 T or CD4 conv cells as comparison that did not show up CD177 staining. Those cells should show similar CD177 positivity if doublet is the cause. Neutrophils exhibit a high level of SSC and the chance of contamination is very low. Rudensky lab has reported CD177 expression on Treg before with no functional elaboration (4).

4. The Treg suppression assays have not been performed correctly. One NEVER adds IL-2 to a Treg suppression assay as studies performed by the Sakaguchi/Shevach labs more than 2 decades ago clearly demonstrated that the addition of IL-2 abrogates or masks the suppressor function of Tregs. Again all the populations used in Fig. 5c must be stained for Foxp3 expression. The authors characterize these populations in the figure legend as Foxp3+, but never show this data and expression of CD25 and Cd127 as pointed out above are not equivalent. Why don't the total Treg population suppress? Are the Treg populations used non-responsive to either TCR or IL-2 stimulation as is characteristic of Foxp3+ Treg in both mouse and man?

Response: we have to apologize for the misleading experimental section. We are well aware the standard suppression initially defined from Sakaguchi/Shevach labs and IL-2 should be excluded. Fig: 5: We have to pre-activate CD4+CD25- naïve T cells from PBMC O/N using anti-CD3, anti-CD28 (dynabeads) and IL-2, since all PBMCs were frozen in liquid nitrogen. These cells were then counted, CFSE labeled, and co-cultured with flow sorted Tregs at different ratios, only using anti-CD3, anti-CD28 dynabeads without IL-2. We clarified in the method section.

We also tested the recombinant CD177-FC, purified from HEK293T cells, to repeat the suppression assay and found that CD177 protein is sufficient to inhibit effector T cell proliferation (Supplementary Figure 5B-E), irrelevant of IL-2 presence. This is consistent with the antibody blocking experiments (Figure 5C-D).

Foxp3 is an intracellular staining and cannot be used for functional assay. We used CD4+CD25+Foxp3+ to define tumor-infiltrating Tregs that are now included as Figure 4D and supplementary Figure 4G. The original Figure 4C was moved to Supple Figure 4H. Again, CD177 is only restricted to be expressed to TI-Tregs defined by either FoxP3+ or by CD25+CD127low.

We did several experiments to show that the CD25+CD127low population exclusively expresses Foxp3+ in several specimens. We used the most popular clone for FoxP3 staining in human (Clone: 236A/E7), but the shift is not as big as we have seen in mouse FoxP3 staining (Supplementary Figure 7B).

Below is an example showing TI-Tregs (CD4+CD25+CD127low) express FoxP3 but not CD4+CD25- Tconv cells.

For the suppressive capacity of total TI-Tregs, we totally agree that these cells should have suppressive function. Please note we are using human cancer Tregs in a classic suppression setting. The number of these TI-Tregs are very limited and the suppression assay requires certain numbers/concentrations. We cannot purify unlimited TI-Tregs to do these experiments. Fig.5B is from 3 largest tumors we got and Fig 5C-D is from 5 largest tumors we got (>500mg a piece). The large size of tumors in the clinical setting are extremely rare and it took us multiple years to accumulate enough specimens. We barely have enough cells to do what we showed in these Figures. All single cells from these specimens had to be frozen/thawed for combination and flow sorting. The suppressive capacity is nowhere close to freshly isolated Tregs as in mice or in human PBMC. We showed 1:2 Teff/Treg ratio in Fig5B-D to show the best suppression only from CD177+ Tregs, but total Tregs only have less than 20% CD177+ Tregs (**Figure. 4E**). We don't have enough total Tregs to do the experiment, but we believe if we increase to 5 times total TI-Tregs, we should see a similar suppression.

5. The reversal of suppression by the anti-CD177 is a very intriguing observation as it implies that CD177 is actually involved in the in vitro function of Tregs. This claim requires much greater experimentation. First, the Treg suppression assay needs to be performed in the absence of IL-2 using standard conditions. Secondly, CD177 is a member of the Ly-6 family of antigens and antibodies to this family of antigens in either mouse or man (e.g, CD59, JI 146, 4092, 1991) have potent T cell activating properties under certain conditions in vitro. The authors must prove that the Treg is the actual target for the effects of the mAb and not the responding T cell. What cells are actually dividing in Fig. 5D?

Response:

First: Yes, we did suppression assay without IL-2 as explained before and updated in the methods.

Second: We also treated the PBMC naive CD4 or CD8 T cells with the same antibody, and we did not find increase/decrease in proliferation from these cells, suggesting that the antibody-mediated T cell proliferation is not working on effector T cells. We do believe anti-CD177 antibody works on inhibition of Treg suppressive function and the antibody does not induce the CD4/8 proliferation.

Here is the data and we can include if the reviewer suggests so. Thanks.

Legend: CD4+CD25- or CD8+ T cells were purified from PBMC, labeled with CFSE, reactivated with anti-CD3/anti-CD28 Dynabeads (Thermofisher). Control, mIgG isotype, or anti-CD177 (MEM166) were incubated at different time points. Proliferation (% of CFSE-diluted cells) is shown.

Fig.5B-D are essentially the same experiments with only difference of with (Figure 5C-D) or without anti-CD177 antibody (Figure 5B). Only Naïve CD4 T cells were labeled by CFSE and can be serially diluted as per cycle of proliferation. Tregs are known to be poorly proliferated with only CD3/CD28 stimulation (9) compared to naïve T cells, and won't show signs of serial dilution (different peaks in Fig.5B suggesting that only naïve T cell – labeled with CFSE can exhibit different peaks representing serial CSFE fluorescence signal).

6. The studies using global and Treg conditional knock out mice are incomplete. The phenotype of both of these mice must be completely characterized. It remains unclear to me if any mouse T cell expresses CD177. Sup Fig 7B is described in the text at showing mouse splenocytes yet the figure and the legend indicate human splenocytes. There are potentially numerous explanations why CD177 deficient Treg fail to suppress in the mouse tumor model. For example, one study has suggested that PECAM-1 is actually the ligand for CD177 and it is therefore possible that the CD177 deficient Tregs fail to enter the tumor microenvironment and thus fail to enhance tumor growth. A complete work up of Treg function in these mice is required before any conclusions can be drawn from the data presented. Commercial mAbs to mouse CD177 are available. It is not clear what mAb is used to identify mouse CD177 in this study.

Response: These are great points. The mouse reagents are very limited and all good antibodies are only for human CD177 and don't cross-react with mouse CD177. During the revision, we were able to identify a newly available monoclonal anti-mouse CD177 antibody that is able to identify mouse CD177 by flow cytometry. We updated the detection of CD177⁺ TI-Tregs within MC38 tumors in WT mice (CD177fl/fl) that were significantly reduced when tumors are from CD177fl/fl/FoxP3-YFP-iCre mice (Jax: 016959) (**Figure 6D**), though not reaching 100% depletion likely due to the inefficiency of Foxp3-cre-YFP or the inefficiency of the stop codon generated by linking exon 1 and exon 3 (the one stop codon generated

after deleting exon 2 may not be strong enough to stop protein expression). In a separate study, we compared CD177 expression within neutrophils from CD177fl/fl or CD177fl/fl/LysM-Cre mice and found efficient deletion of CD177 in Ly6G⁺ neutrophils (**left Figure**).

Legend: Blood cells from CD177fl/fl (WT, blue color) or CD177fl/fl/LysM-Cre (KO, red color) mice were stained for flow

cytometry. Representative picture (left) and statistics (right) were shown.

Mouse splenic Tregs have been shown to express mRNA of CD177 before (10). We were able to detect CD177 on mouse splenic Tregs and thymic Tregs, which is also depleted in Treg-specific KO mice (Supplementary Figure 7C). We also found low levels of CD177⁺ Treg cells in human spleen (Supplementary Figure 4F, those are patients with cancer, we cannot get spleens without cancer) as well, <2% of total splenic Tregs in one patient and ~11% in another (newly updated Supplementary Figure 4F). We do apologize for the initial wrong citation in the text and lack of mentioning human splenic data.

We did more thorough characterization of germline KO by histopathology. We published whole body necropsy in young mice (7-8 weeks old) and did not find any pathological alterations – evaluated by Dr. Gibson-Corley, a comparative pathologist (11); we did more thorough necropsy on aged mice for potential impact of CD177 deficiency on aged mice most over 18 months. Based on the professional evaluations of most essential tissues in the body, Dr. Gibson-Corley concludes that there is no defined autoimmunity or any KO-specific pathological alterations at tissue levels (WT and KO, 3 females and 3 males each). Formal report is included in the source data.

We performed a full immunophenotyping (Supplementary Figure 7B, gating scheme) for CD177^{fl/fl}/FOXP3-cre mice without (Supplementary Figure 8) or with tumors (Supplementary Figure 9), using a complex 21-color flow cytometry. We repeated Figure 6C with two independent experiments and found the same results of reduced tumor growth within Treg-specific CD177 KO mice (right panel). We don't have CD177 germline KO mice in active breeding and cannot perform similar experiments.

While we don't see significant alterations in immune components from non-tumor bearing mice, we did repeatedly observe reduced TI-Tregs from MC38 tumors from the Treg-specific KO mice (Figure 6E, combined from 2 separate experiments for TI-Tregs), correlating with reduced proliferation within TI-Tregs (Figure 6F). This is in agreement with known literature showing that CD177⁺ TI-Tregs have increased clonality, indicative of the proliferative expansion of certain TI-Treg clones (4). There is no significant change in chemokine receptors on splenic Tregs or tumoral Tregs, indicating that the decreased TI-Tregs from Treg-specific KO mice may be a cause of proliferation, a known feature of TI-Tregs (4) or reduced tumor-specific production of thymic Tregs (Figure 6E, T-Thymus). We still cannot rule out the involvement of Treg trafficking that may contribute to TI-Tregs. This phenotype – as indicated from human study – is tumor-specific phenotype since there is no significant difference in Tregs from other tissues except a slight reduction of thymic Tregs from tumor-bearing Treg-specific KO mice (Figure 6E), in agreement with the moderate expression of CD177 on thymic Tregs (Supplementary Figure 7C).

To better understand CD177 deletion at functional genomics, we performed RNAseq of splenic Tregs from tumor bearing spleens or TI-Tregs purified from MC38 tumors, either from CD177^{fl/fl}/FOXP3-cre or CD177^{fl/fl}/WT mice by sorting out CD3⁺CD4⁺CD25⁺GITR⁺ Tregs. Further consolidated by human CD177⁺ versus CD177⁻ TI-Tregs, we did not see big difference in gene expression between WT and CD177-KO Tregs nor pathway difference in known Treg-suppressive gene signature or chemokine receptors that mediate its tumor infiltration (Supplementary Figure 9G-J).

Based on these mouse experiments, and the functional experiments that human CD177⁺ Tregs and WT Tregs are more suppressive than their negative counterparts, we believe that CD177 itself may have immune suppressive function and indeed CD177 protein was able to inhibit effector T cells from human blood (Supplementary Figure 5B-E). We are investigating potential receptor on effector T cells, but we do not have a concrete target yet. As the reviewer suggested, PECAM-1 can be a valid target for Treg cells entering tumor microenvironment but cannot explain the suppressive phenotype for CD177⁺ Tregs. We did not identify any PECAM-1 express on effector T cells.

Overall, we have a lot negative results related to the mechanistic explanation of CD177⁺ Tregs, both from human and mouse TI-Tregs, but our data do support that important role of CD177⁺ Tregs in immune suppression and it can be a good target since its expression on PBMC Tregs is 0% and very low in spleen or normal tissues. Technical limitations (number of TI-Tregs from human cancer specimens and unavailability of anti-mouse CD177 antibody) and the COVID-19 pandemic prevent us from getting more fresh tissues.

Reviewer #2 (Treg, TCR signaling) (Remarks to the Author):

Here, the authors analyze gene expression signatures of individual Treg isolated from human tumors and peripheral blood. They identify sets of up-regulated and down-regulated genes in comparative analysis of TIL and PBL Treg, consistent with previously published data sets of tumors of different origin. The finding that CD177 identifies a particularly suppressive subset of Treg is interesting, as are the results of knocking out CD177, either constitutively or inducibly.

However, it is not clear from the manuscript why the authors chose to focus on CD177 (other than stating later that it was upregulated in a subset of Cluster 1).

Response: The focus on CD177 Treg is historical in the lab. We initially started two separate projects:

1) to study the role of immune modulators on cancer where we found CD177 to be the good (correct, it is a good predictor) for breast cancer prognosis. We followed up this study and found out CD177 is expressed by normal epithelial cells from various tissues including mammary epithelial cells (1), prostate epithelial cells, colonic epithelial cells etc. (Supplementary Figure 4D-E). We found CD177 to be a tumor suppressor due to its cancer/epithelial cell-intrinsic expression and published the work a few month ago (1).

2) to profile immune cell infiltrates from cancers. We focus on ccRCC due to its specific properties in terms of immunotherapy: low mutational burden but very responsive to immunotherapy. We want to know whether immune cell infiltrates may explain why ccRCC is different from classic cancer types (melanoma, lung adeno etc.) for cancer immunotherapy (submitted, and preprint: <https://www.biorxiv.org/content/10.1101/824482v1>).

When we analyzed TI-Tregs, CD177 stands out within the common upregulated genes among ccRCC, HCC, breast cancer, colon cancer, and lung cancers). We confirmed cancers have CD177⁺ lymphocytes based on IHC staining and flow cytometry. Among all the DEGs between TI- and PB-Tregs, CD177 presents on cell surface that can be easily targeted by antibodies and we have developed KO and floxed allele for CD177 as well. These important observations and availability of reagents/mouse models lead us to focus on CD177 rather than other DEGs discovered in Fate-1 Tregs.

It is also not clear from the manuscript how the authors think CD177 may be acting to modify Treg function (i.e. apparently increasing their suppressive activity).

Response:

We responded to reviewer 1, critique 6 for the potential mechanisms how CD177 may modify Treg function. We believe CD177 protein has immune suppressive function, which likely have its own partner on effector T cells for suppression (Supplementary Figure 5B-E). We are actively looking for its interaction proteins on effector T cells and with a few potential hits, but nothing concrete yet. We added more discussion as below:

“We have tried many experiments to understand why CD177+ TI-Tregs are more immune suppressive, including genetic analysis of CD177+ and CD177- Tregs from mouse and human (no significant identifiable change in known suppressive genes) and expression of cell surface proteins (Figure 5). Increased expression of immune checkpoints such as PD-1 and CTLA4 (Figure 5) cannot explain the suppressive effect in vitro. It is well-accepted that Tregs suppress effector T cells via direct contact in vitro (9, 12-15), which drives us to believe the presence of CD177 receptor on effector T cells. We have failed to detect strong expression of known CD177 receptors such as CD31/PECAM-1 and proteinase 3 on effector T cells, our future work is to actively identify the potential CD177-binding receptors on effector T cells.”

Regarding the single cell RNAseq analysis, the authors do not report how many Treg cells are being analyzed in each case, or in total (what is reported is number of immune cells and not the number of Treg). This should be clarified.

Response: We have clarified the numbers in each group in the figure legends for different comparisons (new Figure 4A for ccRCC and Supplementary Figure 4A for HCC).

In Fig. 2, the authors, based on pseudotime analysis, discuss the origination of two clusters of Treg. However, they do not mention that some junctional zone and liver Treg have a similar expression profile (Supplementary Fig. 2A). Thus, the authors should address the fact that these variables may be a common of Treg in non-immune tissues in general, and may not be specific to tumors.

Response: This is indeed correct. There is expression of CD177 in ~10% of normal Liver Tregs. This is consistent with our histological findings in normal tissues in supplemental Figure 4 D-E, as well as in Tregs from normal breast parenchyma (4). However, within the junctional-zone and tumor-infiltrating Treg populations, there is a greater percentage of CD177⁺ Tregs, as well as an upregulation of the mRNA (Supplementary Figure 4A-C). The interesting finding that junctional-zone has increased % of CD177⁺ TI-Tregs supports the notion that certain immune suppression within cancer is located at peritumor area/junctional zone. We also saw CD177⁺ Tregs in human spleen from cancer patients (Supplementary Figure 4F) and mouse thymic Tregs (Supplementary Figure 7C, upper panels). We revised the manuscript to state that the expression of CD177 on Tregs is not restricted to TI-Tregs, but it is induced by tumors (Fig. 6E) and can be a good therapeutic target for cancer therapy.

In Fig. 2 D-E and Supplementary Fig. 2D-E, the authors imply that cluster 1 and 2 have an exhaustion signature. However, it is not clear which genes are being used to define these cells, nor what the cytotoxicity, cell cycle and naïve status signature scores are based on. Of note, “exhaustion” of Treg is not well-defined, particularly at a functional level.

Response: We have added additional text to the methods section to describe the enrichment scores being based on single-sample gene set enrichment using the signatures defined by the singleR package with listed genes for defining exhaustion and other phenotypes. We have clarified that the enrichment

of the exhaustion gene set could be a product of the higher expression of immune checkpoints, that may or may not act as effector molecules for Tregs; hence the exhaustion signature for TI-Tregs – some known in the literature – is correlated with increased suppressive function of Tregs with both clinical and preclinical evidence targeting these exhaustion molecules may decrease Treg cell numbers and/or their suppressive function (16-24). This is also true in breast cancers that metastasis-promoting Tregs have increased levels of PD-1, CTLA-4, and ICOS (8).

In Fig. 4A, the authors should clarify whether the tissue-infiltrating cells from liver show any expression of CD177; also, the status of CD177 in tumor-associated tissue used in the previous figure is not mentioned.

Response: As seen in the new supplementary Figure 4A-C, a similar distribution of increased CD177⁺ Tregs were seen in CF1 within tissue-infiltrating cells or junctional zone Tregs from the liver, relative to CF2 and peripheral T cells. But the expression levels of CD177/% of CD177⁺ Tregs in normal liver tissue is much lower than those from junctional zone and tumors.

In 4C, the authors label CD127 on both the X and Y axes. I think the Y axis should probably be CD177.

Response: it should be CD25 and updated in the new figure (new Supplementary Figure 4H).

In Supplementary Fig. 4D, where the authors show staining and gating controls for Treg analysis, there seems to be a nonspecific shift of the whole CD3 population, compared to the FMO control. Moreover, the sequence of flow analysis in Supp. Fig 4D is odd, where FoxP3 has been used to gate before gating on CD4. It would be better to perform the gating the other way around, i.e. first gate on CD4, then show FoxP3, etc.

Response: Thanks for the suggestions. We re-gated different immune cells, following lymphocyte gating (very stringent gating using lymphocytes from PBMCs or splenocytes, Figure 4D)-single cells (FSC-H/FSC-A)-single cells (SSC-H/SSC-A)-live CD45 immune cells-CD3 T cells-CD4/CD8-CD25/FOXP3-CD177. We performed flow cytometry from frozen single cell suspensions of 5 more independent breast cancer specimens and all samples contained CD177⁺ Tregs defined by CD25⁺FOXP3⁺ (Figure 4D, Supplementary Figure 4G, showing 3 samples).

Fig. 5 and the supporting supplementary figure state are meant to show that CD177 positive Treg do show upregulation of other markers, like PD-1 and CTLA-4. Have the authors looked to see if other molecules associated with potent Treg suppression, like some cytokines and granzyme B, are upregulated in their staining?

Response: We ran out of frozen stocks of single cell suspension of cancer specimens for protein assay and COVID-19 prevents us from getting new fresh cancer specimens due to local regulations. But we did thorough analysis of RNA data, including purified CD177⁺ and CD177⁻ TI-Tregs from 5 breast cancer patients, ccRCC single cell RNAseq data to retrieve CD177⁺ and CD177⁻ TI-Tregs. We compared all Treg-relevant suppressive genes and chemokine receptors and did not find any significant and consistent change between CD177⁺ and CD177⁻ TI-Tregs (New Supplementary Figure 9G-J).

In Fig. 6, the authors use mice with global or Treg-specific KO of CD177. However, they do repeat the in vitro suppression assay with TIL Treg from either one of these mouse models. In addition, they do not perform any lymphocyte characterization (including of Treg), before proceeding to the tumor models. The authors need to address directly whether the global or inducible KO of CD177 leads to any baseline

changes in lymphoid organs. Also, the authors fail to disclose which FoxP3-Cre they are using in their Treg CD177 KO mouse. In addition, in the tumor experiments there is no detailed TIL characterization to say whether total or Treg-specific KO of CD177 leads to any change in various lymphocyte compartment inside the tumors.

Response: please refer to the detailed explanation under reviewer 1, critique 6. We have done thorough characterization of the mouse now (new Figure 6, New Supplementary Figure 7-9). We used FoxP3-YFP-iCre mice (Jax: 016959). Although we did not observe a complete CD177 deletion in TI-Tregs using this Cre, thymic Tregs in the same mice have very efficient deletion of CD177, as well as neutrophils when CD177^{fl/fl} mice were crossed with LysM-Cre (Figure is under reviewer 1, critique 6). The suppression assay is also done (Supplementary Figure 9F).

Finally, since the authors have access to a CD177 blocking antibody, they could test it in the tumor models with WT mice, although I would consider this experiment a lower priority in the context of other issues that should be addressed, as discussed above.

Response: There is no commercially available antibody that inhibits the suppressive function of mouse Tregs. We tested all antibodies targeting mouse CD177 but can only identify one for IHC and one for flow cytometry. We did not find any blocking or neutralizing antibody.

References:

1. Kluz PN, Kolb R, Xie Q, Borcherdig N, Liu Q, Luo Y, Kim MC, Wang L, Zhang Y, Li W, Stipp C, Gibson-Corley KN, Zhao C, Qi HH, Bellizzi A, Tao AW, Sugg S, Weigel RJ, Zhou D, Shen X, Zhang W. Cancer cell-intrinsic function of CD177 in attenuating beta-catenin signaling. *Oncogene*. 2020. Epub 2020/02/12. doi: 10.1038/s41388-020-1203-x. PubMed PMID: 32042113.
2. Dalerba P, Kalisky T, Sahoo D, Rajendran PS, Rothenberg ME, Leyrat AA, Sim S, Okamoto J, Johnston DM, Qian D, Zabala M, Bueno J, Neff NF, Wang J, Shelton AA, Visser B, Hisamori S, Shimono Y, van de Wetering M, Clevers H, Clarke MF, Quake SR. Single-cell dissection of transcriptional heterogeneity in human colon tumors. *Nat Biotechnol*. 2011;29(12):1120-7. Epub 2011/11/15. doi: 10.1038/nbt.2038. PubMed PMID: 22081019; PMCID: PMC3237928.
3. Toyoda T, Tsukamoto T, Yamamoto M, Ban H, Saito N, Takasu S, Shi L, Saito A, Ito S, Yamamura Y, Nishikawa A, Ogawa K, Tanaka T, Tatematsu M. Gene expression analysis of a Helicobacter pylori-infected and high-salt diet-treated mouse gastric tumor model: identification of CD177 as a novel prognostic factor in patients with gastric cancer. *BMC Gastroenterol*. 2013;13:122. Epub 2013/08/01. doi: 10.1186/1471-230x-13-122. PubMed PMID: 23899160; PMCID: PMC3734037.
4. Plitas G, Konopacki C, Wu K, Bos PD, Morrow M, Putintseva EV, Chudakov DM, Rudensky AY. Regulatory T Cells Exhibit Distinct Features in Human Breast Cancer. *Immunity*. 2016;45(5):1122-34. doi: 10.1016/j.immuni.2016.10.032. PubMed PMID: 27851913.
5. De Simone M, Arrigoni A, Rossetti G, Gruarin P, Ranzani V, Politano C, Bonnal RJP, Provasi E, Sarnicola ML, Panzeri I, Moro M, Crosti M, Mazzara S, Vaira V, Bosari S, Palleschi A, Santambrogio L, Bovo G, Zucchini N, Totis M, Gianotti L, Cesana G, Perego RA, Maroni N, Pisani Ceretti A, Opocher E, De Francesco R, Geginat J, Stunnenberg HG, Abrignani S, Pagani M. Transcriptional Landscape of Human Tissue Lymphocytes Unveils Uniqueness of Tumor-Infiltrating T Regulatory Cells. *Immunity*. 2016;45(5):1135-47. Epub 2016/11/17. doi: 10.1016/j.immuni.2016.10.021. PubMed PMID: 27851914; PMCID: PMC5119953.
6. Delgoffe GM, Woo SR, Turnis ME, Gravano DM, Guy C, Overacre AE, Bettini ML, Vogel P, Finkelstein D, Bonnevier J, Workman CJ, Vignali DA. Stability and function of regulatory T cells is

maintained by a neuropilin-1-semaphorin-4a axis. *Nature*. 2013;501(7466):252-6. Epub 2013/08/06. doi: 10.1038/nature12428. PubMed PMID: 23913274; PMCID: PMC3867145.

7. Ha D, Tanaka A, Kibayashi T, Tanemura A, Sugiyama D, Wing JB, Lim EL, Teng KWW, Adeegbe D, Newell EW, Katayama I, Nishikawa H, Sakaguchi S. Differential control of human Treg and effector T cells in tumor immunity by Fc-engineered anti-CTLA-4 antibody. *Proceedings of the National Academy of Sciences of the United States of America*. 2019;116(2):609-18. Epub 2018/12/28. doi: 10.1073/pnas.1812186116. PubMed PMID: 30587582; PMCID: PMC6329979.

8. Nunez NG, Tosello Boari J, Ramos RN, Richer W, Cagnard N, Anderfuhren CD, Niborski LL, Bigot J, Meseure D, De La Rochere P, Milder M, Viel S, Loirat D, Perol L, Vincent-Salomon A, Sastre-Garau X, Burkhard B, Sedlik C, Lantz O, Amigorena S, Piaggio E. Tumor invasion in draining lymph nodes is associated with Treg accumulation in breast cancer patients. *Nat Commun*. 2020;11(1):3272. Epub 2020/07/01. doi: 10.1038/s41467-020-17046-2. PubMed PMID: 32601304; PMCID: PMC7324591.

9. Thornton AM, Shevach EM. CD4+CD25+ immunoregulatory T cells suppress polyclonal T cell activation in vitro by inhibiting interleukin 2 production. *J Exp Med*. 1998;188(2):287-96. Epub 1998/07/22. doi: 10.1084/jem.188.2.287. PubMed PMID: 9670041; PMCID: PMC2212461.

10. Zemmour D, Zilionis R, Kiner E, Klein AM, Mathis D, Benoist C. Single-cell gene expression reveals a landscape of regulatory T cell phenotypes shaped by the TCR. *Nat Immunol*. 2018;19(3):291-301. Epub 2018/02/13. doi: 10.1038/s41590-018-0051-0. PubMed PMID: 29434354; PMCID: PMC6069633.

11. Xie Q, Klesney-Tait J, Keck K, Parlet C, Borchering N, Kolb R, Li W, Tygrett L, Waldschmidt T, Olivier A, Chen S, Liu GH, Li X, Zhang W. Characterization of a novel mouse model with genetic deletion of CD177. *Protein Cell*. 2015;6(2):117-26. Epub 2014/11/02. doi: 10.1007/s13238-014-0109-1. PubMed PMID: 25359465; PMCID: PMC4312768.

12. Takahashi T, Kuniyasu Y, Toda M, Sakaguchi N, Itoh M, Iwata M, Shimizu J, Sakaguchi S. Immunologic self-tolerance maintained by CD25+CD4+ naturally anergic and suppressive T cells: induction of autoimmune disease by breaking their anergic/suppressive state. *Int Immunol*. 1998;10(12):1969-80. Epub 1999/01/14. doi: 10.1093/intimm/10.12.1969. PubMed PMID: 9885918.

13. Piccirillo CA, Shevach EM. Cutting edge: control of CD8+ T cell activation by CD4+CD25+ immunoregulatory cells. *J Immunol*. 2001;167(3):1137-40. Epub 2001/07/24. doi: 10.4049/jimmunol.167.3.1137. PubMed PMID: 11466326.

14. Dieckmann D, Plottner H, Berchtold S, Berger T, Schuler G. Ex vivo isolation and characterization of CD4(+)CD25(+) T cells with regulatory properties from human blood. *J Exp Med*. 2001;193(11):1303-10. Epub 2001/06/08. doi: 10.1084/jem.193.11.1303. PubMed PMID: 11390437; PMCID: PMC2193384.

15. Jonuleit H, Schmitt E, Stassen M, Tuettenberg A, Knop J, Enk AH. Identification and functional characterization of human CD4(+)CD25(+) T cells with regulatory properties isolated from peripheral blood. *J Exp Med*. 2001;193(11):1285-94. Epub 2001/06/08. doi: 10.1084/jem.193.11.1285. PubMed PMID: 11390435; PMCID: PMC2193380.

16. Wing K, Onishi Y, Prieto-Martin P, Yamaguchi T, Miyara M, Fehervari Z, Nomura T, Sakaguchi S. CTLA-4 control over Foxp3+ regulatory T cell function. *Science*. 2008;322(5899):271-5. Epub 2008/10/11. doi: 10.1126/science.1160062. PubMed PMID: 18845758.

17. Paterson AM, Lovitch SB, Sage PT, Juneja VR, Lee Y, Trombley JD, Arancibia-Carcamo CV, Sobel RA, Rudensky AY, Kuchroo VK, Freeman GJ, Sharpe AH. Deletion of CTLA-4 on regulatory T cells during adulthood leads to resistance to autoimmunity. *J Exp Med*. 2015;212(10):1603-21. Epub 2015/09/16. doi: 10.1084/jem.20141030. PubMed PMID: 26371185; PMCID: PMC4577848.

18. Simpson TR, Li F, Montalvo-Ortiz W, Sepulveda MA, Bergerhoff K, Arce F, Roddie C, Henry JY, Yagita H, Wolchok JD, Peggs KS, Ravetch JV, Allison JP, Quezada SA. Fc-dependent depletion of tumor-infiltrating regulatory T cells co-defines the efficacy of anti-CTLA-4 therapy against melanoma. *J Exp Med*. 2013;210(9):1695-710. Epub 2013/07/31. doi: 10.1084/jem.20130579. PubMed PMID: 23897981; PMCID: PMC3754863.

19. Bulliard Y, Jolicoeur R, Windman M, Rue SM, Ettenberg S, Knee DA, Wilson NS, Dranoff G, Brogdon JL. Activating Fc gamma receptors contribute to the antitumor activities of immunoregulatory receptor-targeting antibodies. *J Exp Med*. 2013;210(9):1685-93. Epub 2013/07/31. doi: 10.1084/jem.20130573. PubMed PMID: 23897982; PMCID: PMC3754864.
20. Selby MJ, Engelhardt JJ, Quigley M, Henning KA, Chen T, Srinivasan M, Korman AJ. Anti-CTLA-4 antibodies of IgG2a isotype enhance antitumor activity through reduction of intratumoral regulatory T cells. *Cancer Immunol Res*. 2013;1(1):32-42. Epub 2014/04/30. doi: 10.1158/2326-6066.CIR-13-0013. PubMed PMID: 24777248.
21. Ma Q, Liu J, Wu G, Teng M, Wang S, Cui M, Li Y. Co-expression of LAG3 and TIM3 identifies a potent Treg population that suppresses macrophage functions in colorectal cancer patients. *Clin Exp Pharmacol Physiol*. 2018;45(10):1002-9. Epub 2018/06/16. doi: 10.1111/1440-1681.12992. PubMed PMID: 29905955.
22. Huang CT, Workman CJ, Flies D, Pan X, Marson AL, Zhou G, Hipkiss EL, Ravi S, Kowalski J, Levitsky HI, Powell JD, Pardoll DM, Drake CG, Vignali DA. Role of LAG-3 in regulatory T cells. *Immunity*. 2004;21(4):503-13. Epub 2004/10/16. doi: 10.1016/j.immuni.2004.08.010. PubMed PMID: 15485628.
23. Kurtulus S, Sakuishi K, Ngiow SF, Joller N, Tan DJ, Teng MW, Smyth MJ, Kuchroo VK, Anderson AC. TIGIT predominantly regulates the immune response via regulatory T cells. *The Journal of clinical investigation*. 2015;125(11):4053-62. Epub 2015/09/29. doi: 10.1172/JCI81187. PubMed PMID: 26413872; PMCID: PMC4639980.
24. Giancchetti E, Fierabracci A. Inhibitory Receptors and Pathways of Lymphocytes: The Role of PD-1 in Treg Development and Their Involvement in Autoimmunity Onset and Cancer Progression. *Front Immunol*. 2018;9:2374. Epub 2018/11/06. doi: 10.3389/fimmu.2018.02374. PubMed PMID: 30386337; PMCID: PMC6199356.

REVIEWER COMMENTS

Reviewer #1 (Remarks to the Author):

The expression of CD177 on Treg still remains poorly characterized. The authors wish to claim that it is primarily induced in the tumor microenvironment and is not a marker of activated Treg. Yet, the data presented supports the concept that it is a marker of activated Treg. Most activated Treg markers are elevated on the CD177+ T cells from the tumors. They never perform the simple experiment of activating human Treg from PBMC (no limitation on numbers) and stimulate them for several days with anti-CD3/CD28 beads and high concentrations of IL-2 and determine the expression of CD177. Is the superior suppressor function of this population simply secondary to their activation status which has been shown in numerous papers. Similar activation studies should be done with CD4+ non-Treg to see if CD177 can also be induced on effector cells. It is entirely possible that activated T effectors as used in Fig. 5C and D might express CD177 and that the reversal of suppression is mediated by targeting activated effector cells. As the authors point out, reversal of Treg suppression is a very unusual finding. Many of the claims concerning reversal of suppression by antibodies to members of the TNFSF such as the GITR and OX40 were in the end shown to be directed at enhancing Treg activation. The data presented in Figure 5 (B,C, D) as well as in supplement 5 needs to be greatly expanded.

The authors use a 2 cell suppressor assay in the absence of APC. While a certain degree of suppression may be seen under these conditions, a much more sensitive assay using human non-Treg APC and soluble APC would be preferable. It still remains unclear why total Treg and CD177- Treg produce almost no suppression even at relatively high Treg to responder ratios (1:2).

The significance of the very minor effects of CD177-Fc on effector T cell proliferation are difficult to interpret. Would CD177-Fc block IL-2 mediate proliferation of T cell blasts in the absence of anti-CD3/CD28. Such a study might pin point where the fusion protein acts. It is also curious that the dye dilution curves of the effector cells in the presence of absence of IL-2 are EXACTLY the same and appear to have been duplicated. In general, IL-2 would augment proliferation.

The availability of a Treg specific CD177 mouse affords the author multiple opportunities to test the role of CD177 expression on T cell activation in vitro. Where are these studies? Again, evaluation of the expression of CD177 under multiple activation conditions of mouse Treg should be performed. They now have an antibody to mouse CD177 which appears to stain.

The studies in the mouse tumor model clearly show enhanced tumor immunity in the cKO mouse. The question is why? The most likely explanation from the data shown appears to be a defect in the migration of Treg to the tumor, not a defect in Treg suppression. This needs to be explored further in depth. The differences in thymic generation of Ki-67 in the tumor site do not appear to be biologically significant.

The authors assume that Treg directly interact with CD8+ effector T cells. The site a number of "ancient history" papers from the early 2000s that support this view, but it is far from clear that this is the case. The authors may be searching at the wrong cell type for a purported CD177 ligand.

Reviewer #2 (Remarks to the Author):

The authors have very thoroughly addressed my critiques. I have no additional comments.

Point-by-Point Response

Overall response: We really appreciate the reviewer's comments and critiques that make the paper in a much better shape. We re-arranged some figures and included many new experiments to support our claims. New data include: Fig. 4f; Fig. 4g; Fig. 6b; Fig. 6d; Fig. 6e; Fig. 6f; Fig. 6g; Fig. 6g; Fig. 6i; Fig. 6j; Fig. 7b; Fig 7d; Fig. 7e; Fig. 7f. We also removed some figures based on the reviewer's comments detailed below.

Comment-1: The expression of CD177 on Treg still remains poorly characterized. The authors wish to claim that it is primarily induced in the tumor microenvironment and is not a marker of activated Treg. Yet, the data presented supports the concept that it is a marker of activated Treg. Most activated Treg markers are elevated on the CD177+ T cells from the tumors. They never perform the simple experiment of activating human Treg from PBMC (no limitation on numbers) and stimulate them for several days with anti-CD3/CD28 beads and high concentrations of IL-2 and determine the expression of CD177. Is the superior suppressor function of this population simply secondary to their activation status which has been shown in numerous papers. Similar activation studies should be done with CD4+ non-Treg to see if CD177 can also be induced on effector cells. It is entirely possible that activated T effectors as used in Fig. 5C and D might express CD177 and that the reversal of suppression is mediated by targeting activated effector cells. As the authors point out, reversal of Treg suppression is a very unusual finding. Many of the claims concerning reversal of suppression by antibodies to members of the TNFSF such as the GITR and OX40 were in the end shown to be directed at enhancing Treg activation. The data presented in Figure 5 (B,C, D) as well as in supplement 5 needs to be greatly expanded.

Response: We appreciate the suggestions and performed several experiments to demonstrate that CD177 is not induced by anti-CD3/CD28 beads and high concentrations of IL-2 from Tregs and Tconv cells using purified cells from human PBMC or mouse splenocytes. We now put the new data as **new Fig 4f-g**. In addition, we treated effector CD4 and CD8 T cells with anti-CD177 antibody and did not see the direct impact of the antibody on CD4 or CD8 T cell proliferation (**New Fig. 7d**).

Comment-2: The authors use a 2 cell suppressor assay in the absence of APC. While a certain degree of suppression may be seen under these conditions, a much more sensitive assay using human non-Treg APC and soluble APC would be preferable. It still remains unclear why total Treg and CD177- Treg produce almost no suppression even at relatively high Treg to responder ratios (1:2).

Response: We followed the suggestion and did APC based T cell activation using human monocyte derived dendritic cells (**Fig. 7b**, 5:1 ratio Teff/Treg), or using mouse splenocytes (T cell depleted) for activation of T cells (**Fig. 7e**, 4:1 to 1:1 ratios). We did see CD177- TI-Tregs suppress partially T cell proliferation, even though not as significant as CD177+ TI-Tregs at 5:1 ratio (**Fig. 7b**). These are the combination of 7 large renal cell carcinomas from a collaborator group. Mouse TI-Tregs also exhibited similar pattern and CD177+ TI-Tregs are more suppressive than CD177- TI-Tregs (**Fig. 7e**). The partial suppression of APC-based T cell proliferation by CD177- TI Tregs is likely due to the role of CTLA-4 in competing with CD28 for CD80/CD86 co-stimulation at the priming/activation stage of effector T cells, presumably within secondary lymphoid tissues. The more suppressive effect of CD177+ TI Tregs is likely due to other unknown mechanism within tumor microenvironment, either directly or indirectly, which warrants further investigation. In addition to the figures, we also include the discussion related to this part.

The significance of the very minor effects of CD177-Fc on effector T cell proliferation are difficult to interpret. Would CD177-Fc block IL-2 mediated proliferation of T cell blasts in the absence of anti-CD3/CD28. Such a study might pinpoint where the fusion protein acts. It is also curious that the dye dilution curves of the effector cells in the presence of absence of IL-2 are EXACTLY the same and appear to have been duplicated. In general, IL-2 would augment proliferation.

Response: We agree with the comment that the impact of CD177-FC on T cell proliferation is very minor. We repeated and the trend was the same. Based on the comment here and the discussion between all the authors, we decided to remove the supplementary data to avoid confusion and misleading information.

Comment-3: The availability of a Treg specific CD177 mouse affords the author multiple opportunities to test the role of CD177 expression on T cell activation in vitro. Where are these studies? Again, evaluation of the expression of CD177 under multiple activation conditions of mouse Treg should be performed. They now have an antibody to mouse CD177 which appears to stain.

Response: Thanks for the comments. We did include a lot more experiments based on the suggestions and included Fig. 4g; Fig. 6b, Fig. 6f-j; Fig. 7e-f for the recommended experiments.

Comment-4: The studies in the mouse tumor model clearly show enhanced tumor immunity in the cKO mouse. The question is why? The most likely explanation from the data shown appears to be a defect in the migration of Treg to the tumor, not a defect in Treg suppression. This needs to be explored further in depth. The differences in thymic generation of Ki-67 in the tumor site do not appear to be biologically significant.

Response: We included more data to address the potential impact of CD177 on migration in vivo (**Fig. 6f; Fig. 6h**) and in vitro (**Fig. 6j**), suppression (**Fig. 7b, Fig. 7d, Fig. 7e-f**), proliferation (original figure 5, now **Fig. 7a-b**), and viability (**Fig. 7g; Fig. 7i**) based on the suggestion. We also moved original figures about the chemokine receptors to **Fig. 7d-e**. All our effort concludes that there is no impact of CD177 on Treg tumor recruitment to tumors or viability. There is still possibility of Treg retention due to the expression of chemokine receptors such as CCR8, but there is no good method in our lab or collaborator laboratory to perform Treg retention within tumors.

Related to the Ki-67 positive Tregs, please note it is the percentage of live CD45 cells, which translates into 5-10% proliferation in TI Tregs and is biologically and statistically significant (current **Fig. 6a; Fig. 6c**). Our suppression assay, both CD3/CD28- and APC-based T cell proliferation is strongly inhibited by CD177⁺ TI-Tregs in human and in mouse (**New Fig. 7**). We do believe that CD177⁺ TI-Tregs have superior suppressive capacity than CD177⁻ TI-Tregs, but we can certainly tune down the statement if the reviewer suggests.

Comment-5: The authors assume that Treg directly interact with CD8⁺ effector T cells. The site a number of “ancient history” papers from the early 2000s that support this view, but it is far from clear that this is the case. The authors may be searching at the wrong cell type for a purported CD177 ligand.

Response: We appreciate the comment and the suggestion about the putative potentially risky ligand identification. We removed the CD177-FC data and also tuned down whether CD177⁺ TI Tregs suppress effector T cells via direct or indirect mechanisms. We did search recent literature about TI Tregs with a focus in human. The information related to direct versus indirect suppression mechanisms of TI-Tregs in human is very sparse and most experiments are co-culturing without separation. The major limitation of TI-Treg study is the number of TI-Tregs we can purify from cancer specimens for classic suppression assay, where we normally euthanize tens of tumor bearing mice for one experiment and even harder for

human specimens. Our data agree with the major conclusion from most of the publications that TI Tregs are highly suppressive, but at this stage, we don't know the mechanisms and leave it open in discussion.

REVIEWER COMMENTS

Reviewer #1 (Remarks to the Author):

The authors have addressed many of the major criticisms I raised in my initial review of the paper and the additional data on the expression of CD177 under different situations is quite convincing and supports their contention that CD177 on Tregs is specific to the tumor microenvironment. However, the functional characterization of the purported enhanced suppressive capacity of CD177+ Tregs and the effect of anti-CD177 in reversing suppression are still not optimally executed:

1. In general one does not perform Treg suppression assays by first pre-activating the responder cells with anti-CD3/CD28 and IL-2 for 24 hours before adding Treg. This would make suppression more difficult to observe and likely accounts for the failure to observe suppression by Total Treg and CD177- Tregs in Figure 7A. It is also not clear if IL-2 is maintained in the suppression assay for the entire length of then assay. The addition of IL-2 would mask the suppressive function of Treg.
2. In the anti-CD3 and APC system (figures 7b and c), it is not clear if the responder cells are also pre-activated for 24 hours.
3. The authors quantitate suppression a number of different ways which are not all the same =CFSE measurement of proliferation, total cell yield and percent non proliferating cells. This renders data interpretation difficult.
4. The differences between the CD177+ and CD177- populations could be secondary to minor differences in the purity of the sorted populations for Fozp3+ Treg. This criticism applies to both the mouse and human studies. In the mouse, the authors could have used Treg from Foxp3 reporter mice where purity could be easily monitored.
5. As pointed in my initial review, the reversal of suppression by anti-CD177 is an unusual finding and needs to be explored in greater depth. It appears that Fig. 7c is the result of a single experiment and needs to be repeated with a complete titration of Treg using a CFSE assays with a complete display of the data.
7. Can suppression by mouse CD177+ Treg also be reversed by anti-mouse CD177? Do Treg (from spleen or the TME) from CD177 cKo mice have normal suppressive function?

The authors have addressed many of the major criticisms I raised in my initial review of the paper and the additional data on the expression of CD177 under different situations is quite convincing and supports their contention that CD177 on Tregs is specific to the tumor microenvironment. However, the functional characterization of the purported enhanced suppressive capacity of CD177+ Tregs and the effect of anti-CD177 in reversing suppression are still not optimally executed:

Response: Thanks for the recognition of our effort in resolving the outstanding issues raised by the reviewer. We really appreciate these constructive comments that make our discovery with more solid support.

1. In general, one does not perform Treg suppression assays by first pre-activating the responder cells with anti-CD3/CD28 and IL-2 for 24 hours before adding Treg. This would make suppression more difficult to observe and likely accounts for the failure to observe suppression by Total Treg and CD177- Tregs in Figure 7A. It is also not clear if IL-2 is maintained in the suppression assay for the entire length of then assay. The addition of IL-2 would mask the suppressive function of Treg.

Response: Since most data are clinical specimens and we have to freeze down tumor-infiltrating immune cells in liquid nitrogen. We did find the recovery of effector T cells is critical for the suppression assay to work. We did in general allow them to recover for overnight after purification, when we don't see any CFSE dilution. We will spin down all effector T cells for accurate counting, label them with CFSE, co-culture with flow cytometry-purified Tregs on the 2nd day. We always find these effector T cells proliferative much better and we did remove IL-2 from the medium after combination of effector T cells and Tregs, as we have clarified in the first response letter and the method section.

2. In the anti-CD3 and APC system (Figures 7b and c), it is not clear if the responder cells are also pre-activated for 24 hours.

Response: We performed these experiments after we received the first round of comments and did not include IL-2. We found this APC-based system works very well and the Teff cells did not need to be pre-activated. Thanks for the great suggestion.

3. The authors quantitate suppression a number of different ways which are not all the same CFSE measurement of proliferation, total cell yield and percent non proliferating cells. This renders data interpretation difficult.

Response: We have changed all to percent proliferation in Figure 7, except for Fig.7c where the CFSE labeling did not work. We have to count total Tconv cells.

4. The differences between the CD177+ and CD177- populations could be secondary to minor differences in the purity of the sorted populations for Foxp3⁺ Treg. This criticism applies to both the mouse and human studies. In the mouse, the authors could have used Treg from Foxp3 reporter mice where purity could be easily monitored.

Response: Thanks for the comments. We did perform carefully pre- and post-sorting of CD177+ and CD177- TI Tregs (New Figures: Supplementary Figure 9b-d for human and Fig. 7e upper left pane for mouse Tregs). The % of FOXP3⁺ Tregs is similar between CD177+ and CD177- TI Tregs.

5. As pointed in my initial review, the reversal of suppression by anti-CD177 is an unusual finding and needs to be explored in greater depth. It appears that Fig. 7c is the result of a single experiment and needs to be repeated with a complete titration of Treg using a CFSE assays with a complete display of the data.

6. Can suppression by mouse CD177+ Treg also be reversed by anti-mouse CD177? Do Treg (from spleen or the TME) from CD177 cKO mice have normal suppressive function?

Response to Comment 5-6: We greatly appreciate the recognition of the significance. The human TI-Tregs are extremely rare to perform the requested experiments, as reflected by the rarity of similar TI Treg-suppressive experiments in PUBMED. We have communicated this issue with our handling editor who also believes that it is impossible to do these human experiments.

Please note that the suppressive capacity of CD177⁺ TI-Tregs has been very firmly established, including Figure 7a (one biological repeat from combined 3 breast cancers); Figure 7b (two biological repeats with 5 breast cancer specimens); Figure 7c (3-4 biological repeats with 7 renal cancers). Due to the complex nature of human specimens, we normally process and flow-sort individual human tissues to collect different populations of TI Tregs since one bad tissue with a lot necrosis may influence the productivity of all other tissues. For Figure 7a, we combined all 3 to get enough cells for one biological replicate at each Teff/Treg ratio; we used the data points from Figure 7b (IgG treatment) to get 2 biological replicates for CD177⁻ TI Tregs and 3 biological replicates for CD177⁺ TI Tregs. For Figure 7b, we have two specimens with dominant numbers of CD177⁺ TI-Tregs and the other 3 specimens were combined to the two dominant specimens. Figure 7c, RCC normally are larger tumors and we have 4 out 7 specimens with relatively more CD177⁺ TI-Tregs, with the other 3 combined and distributed into the 4 specimens. Figure 7a, 7b or 7c each represents one separate experiment with various number of replicates.

For animal experiments, we repeated the MC38 tumor growth experiments as shown in Fig 5c using 20 mice in WT and 14 mice in Treg-specific KO group (right Figure, Panel A). We combined 3-4 tumors per group for sorting CD4⁺CD25⁺GITR⁺ TI-Treg cells for suppression assay (Figure 7e at Teff/Treg 2:1 ratio). With so many mice, we have achieved 4 biological replicates. We found there is no difference of splenic Tregs from WT or KO mice without tumors (Supplementary Figure 9i). We have plenty of TI-Tregs from WT control tumors and performed anti-mouse CD177 experiments. The research related to mouse CD177 is just starting and all antibodies have no information related to the blocking function of these antibodies. Our laboratory has led the production of the first CD177 KO strain and the first CD177^{fl/fl} strain. We only found one polyclonal rabbit antibody showing minor blocking efficiency to TI-Tregs (right Figure, Panel B). The difference is too minor to be included in the paper.

For antibody blockage experiments, please instruct what to include or not. We are okay to remove the data in Figure 7b and related antibody blockage experiment overall. The finding of CD177⁺ TI-Tregs is very significant and our discovery should have general impact in cancer Treg research.

REVIEWERS' COMMENTS

Reviewer #1 (Remarks to the Author):

None